# Fast Sampling via De-randomization for Discrete Diffusion Models

## Abstract

Diffusion models have emerged as powerful tools for high-quality data generation, such as image generation. Despite its success in continuous spaces, discrete diffusion models, which apply to domains such as texts and natural languages, remain under-studied and often suffer from slow generation speed. In this paper, we propose a novel de-randomized diffusion process, which leads to an accelerated algorithm for discrete diffusion models. Our technique significantly reduces the number of function evaluations (i.e., calls to the neural network), making the sampling process much faster. Furthermore, we introduce a continuous-time (i.e., infinite-step) sampling algorithm that can provide even better sample qualities than its discrete-time (finite-step) counterpart. Extensive experiments on natural language generation and machine translation tasks demonstrate the superior performance of our method in terms of both generation speed and sample quality over existing methods for discrete diffusion models. [1]

## 1 Introduction

The diffusion-based generative model (Sohl-Dickstein et al., 2015) has demonstrated its powerful ability to generate high-quality samples in the domains including images (Ho et al., 2020; Song & Ermon, 2020), audios (Chen et al., 2020; Kong et al., 2020) and videos (Ho et al., 2022). A diffusion model includes a forward process that gradually corrupts training data into pure noise and a backward/reverse process that decodes the random noises. During the training, the model will learn a neural function by minimizing an objective closely related to maximum likelihood estimation. Given the learned neural network, the model can generate high-quality samples based on different decoding strategies, such as implicit dynamics (Song et al., 2020a), analytical processes (Bao et al., 2022), and differential equation solvers (Song et al., 2020b; Liu et al., 2022; Lu et al., 2022).

While the above diffusion models focus on continuous-state spaces, diffusion models in discrete-state spaces have also gained significant attention for their application in image segmentation (Hoogeboom et al., 2021b), medical record generation (Ceritli et al., 2023; Yuan et al., 2023), and text generation (Hoogeboom et al., 2021b; Austin et al., 2021; Zheng et al., 2023). Inspired by the discrete nature of the data, several studies (Sohl-Dickstein et al., 2015; Hoogeboom et al., 2021b; Austin et al., 2021) have introduced discrete diffusion models, which operate directly on discrete state spaces. However, in contrast to their continuous counterparts, discrete diffusion models remain relatively less explored, particularly on the aspect of sampling efficiency. In order to accelerate discrete diffusion models, Zheng et al. (2023) recently proposed a model based on a novel reparameterization technique. The resulting reparameterized diffusion model (RDM) significantly improves the sampling speed compared with vanilla discrete diffusion models (Hoogeboom et al., 2021b; Austin et al., 2021) and achieves better performance than these baselines on text generation tasks.

Unlike the continuous diffusion model, which employs a Gaussian distribution, the discrete diffusion model uses categorical white noises to corrupt the data. By analyzing this special formula, we introduce a de-randomized generation process with an accelerated algorithm. Compared with existing discrete diffusion models, the improved algorithm accelerates sampling under the same number of sampling steps. Notably, this new sampling technique does not necessitate any changes to the training objective of diffusion models. As a result, it can be applied to any discrete diffusion model as long as we can define a de-randomized reverse process and its corresponding accelerated algorithm. Our main contributions are summarized as follows:

---

[1]Our implementation can be accessed via https://anonymous.4open.science/r/DNDM.

- We propose a discrete non-Markov diffusion model (DNDM) by introducing a set of latent variables $\mathcal{T}$, termed *transition time set*. For the original diffusion trajectory $\{\mathbf{x}_t\}$ where $\mathbf{x}_0$ denotes the real example, the new process provably preserves the distribution of $q(\mathbf{x}_t)$ as well as the conditional distribution $q(\mathbf{x}_0|\mathbf{x}_t)$. Since $q(\mathbf{x}_t)$ of the new diffusion process is identical to that of the exact process, the new process can be used as an alternative for diffusion and data generation. Since $q(\mathbf{x}_0|\mathbf{x}_t)$ remains the same, we can choose from a large family of generative models using the same neural network.
- Leveraging the de-randomized diffusion process, we propose an accelerated algorithm, drastically reducing the frequency of neural network calls. In a $T$ time-step discrete diffusion process, where the original sampling (Ho et al., 2020; Austin et al., 2021) need to invoke the neural network $T$ times, our approach requires only $K$ calls (where $K$ represents the cardinality of *transition times* $\mathcal{T}$). More precisely, $K$ is less than $T$ and approaches $O(1)$ as $T$ goes to infinity. Therefore, our algorithm is faster, and the improvement becomes more significant as $T$ increases. Notably, our algorithm on machine translation benchmarks is about $3\times$ faster than RDM (Zheng et al., 2023) for $T = 50$ and about $30\times$ faster for $T = 1000$ while preserving the sample quality.
- To further illustrate the effectiveness of the de-randomized diffusion process, we explore the limit as $T \to \infty$ and introduce an infinite-step sampling algorithm. With a pretrained neural network, we can generate an initial noise $\mathbf{x}_T$ and a transition time set $\mathcal{T} \subset [0, 1]$ with infinitesimal spacing, such that $|\mathcal{T}| = O(1)$. This enables the generation of the real data distribution with only $|\mathcal{T}|$ neural network evaluations. Experiments show that this infinite-step approach yields a superior sample quality compared to finite-step methods.

**Notation.** We use lowercase letters to denote scalars, boldface lowercase letters for vectors, and boldface uppercase letters for matrices. Consider a sequence $x_k$. If $x_k = \mathcal{O}(y_k)$, this implies that there exists a positive constant $C$ such that $|x_k| \le C|y_k|$. The notation $1 : N$ indicates the sequence from $1$ through $N$. The symbol $\mathbf{q}$ designates the real distribution in a diffusion process, while $\mathbf{p}$ represents the distribution during sampling. With its success probability inside the parentheses, the Bernoulli distribution is denoted as $\mathrm{Bernoulli}(\cdot)$. We further use $\mathrm{Cat}(\mathbf{p})$ to denote a categorical distribution over a one-hot row vector with probabilities given by the row vector $\mathbf{p}$.

## 2 DE-RANDOMIZED DIFFUSION PROCESS

In this section, we will first provide the background of the discrete Markov diffusion model, and then introduce our discrete non-Markov diffusion model (DNDM).

### 2.1 DISCRETE MARKOV PROCESS

We begin by briefly describing the discrete Markov diffusion model, which is defined on categorical random variables. Let $\mathbf{x} \in \mathbb{R}^K$ represents a one-hot encoded discrete random variable with $K$ categories. More specifically, if $\mathbf{x}$ equals the standard basis $\mathbf{e}_k$, then the category of $\mathbf{x}$ is $k$. We use $\{\mathbf{x}_t\}_{t=0}^{T}$ to denote the process of random variables across time $0 \le t \le T$, where $\mathbf{x}_0 \in \mathbb{R}^K$ represents data drawn from the real distribution $\mathbf{q}_{\mathrm{data}}$. Discrete diffusion probabilistic models (Hoogeboom et al., 2021b; Austin et al., 2021) are characterized by both forward and backward processes. The forward process aims to gradually transform the real distribution $\mathbf{q}_{\mathrm{data}}$ to some noise distribution $\mathbf{q}_{\mathrm{noise}}$ through $T$ intermediate latent variables $\mathbf{x}_1, \ldots \mathbf{x}_T$ with the following process:

$$\mathbf{x}_t = b_t \mathbf{x}_{t-1} + (1 - b_t)\mathbf{w}_t, \tag{1}$$

where $b_t$ are independently drawn from the Bernoulli distribution $\mathrm{Bernoulli}(\beta_t)$, $\mathbf{w}_t$ are independently drawn from the noise distribution $\mathbf{q}_{\mathrm{noise}}$, which can be a uniform noise distribution over the vocabulary $\{1, 2, \ldots, K\}$ (Hoogeboom et al., 2021b) or a point mass with all of the probability on an absorbing state (Austin et al., 2021). The process in (1) defines a Markov process characterized by the transition kernel

$$q(\mathbf{x}_t|\mathbf{x}_{t-1}) = \mathrm{Cat}\big(\beta_t \mathbf{x}_{t-1} + (1 - \beta_t)\mathbf{q}_{\mathrm{noise}}\big). \tag{2}$$

The property of Markov chain enables us to get samples $\mathbf{x}_{0:t}$ by $p(\mathbf{x}_{1:t} \mid \mathbf{x}_0) = \prod_{i=1}^{t} q(\mathbf{x}_t \mid \mathbf{x}_{t-1})$. Marginalizing over $p(\mathbf{x}_{1:t} \mid \mathbf{x}_0)$ further allows us to sample $\mathbf{x}_t$ for arbitrary time $t$ directly from $\mathbf{x}_0$:

$$q(\mathbf{x}_t|\mathbf{x}_0) = \mathrm{Cat}\big(\alpha_t \mathbf{x}_0 + (1 - \alpha_t)\mathbf{q}_{\mathrm{noise}}\big), \tag{3}$$

where $\alpha_t := \Pi_{s=1}^{t}\beta_s$ is determined by the sequence of $\beta_t$ of our choice and decreases from 1 to 0 as $t$ goes from 1 to $T$.

**Training the Neural Networks.** During training, discrete diffusion algorithms aim to train a function capable of predicting $\mathbf{x}_0$ given $\mathbf{x}_t$. Specifically, they strive to learn a neural network $f_\theta(\cdot, t)$ such that $f_\theta(\mathbf{x}_t, t)$ predicts $\mathbf{x}_0$. The training process typically involves the subsequent steps (Hoogeboom et al., 2021b; Austin et al., 2021; Zheng et al., 2023):

- Generate pairs $(\mathbf{x}_0, \mathbf{x}_t)$ according to the joint distribution $q(\mathbf{x}_0, \mathbf{x}_t)$ stemming from the process in (1), aggregated into the training data set.
- Define a loss objective for the training set at the $t$-th step, denoted by $L_t(f_\theta(\cdot, t), q(\mathbf{x}_0, \mathbf{x}_t))$. Then the training objective is to minimize the average loss given by $\arg\min_\theta (1/T) \sum_{t=0}^{T} L_t(f_\theta(\cdot, t), q(\mathbf{x}_0, \mathbf{x}_t))$.

The formulation of the loss $L_t$ differs across methods. Hoogeboom et al. (2021b) utilized $L_t$ derived from the negative variational bound, which corresponds to the KL divergence between two categorical distributions. Building on this foundation, Austin et al. (2021) introduced an auxiliary denoising objective, which aims at refining the predictions of data $\mathbf{x}_0$ at each time step. Furthering the advancements, Zheng et al. (2023) put forth a reparametrized loss $L_t$ that incorporates a reweighted parameter $\lambda_t$, enhancing its trainability. While our research primarily revolves around reverse sampling, we've relegated detailed discussions of these losses to Appendix B. Importantly, our DNDM can be seamlessly integrated with the neural function $f_\theta(\cdot, t)$ as guided by the aforementioned losses.

**Reverse Sampling.** Given the forward Markov process, the reverse process can be derived by Bayes' rule (Hoogeboom et al., 2021b; Austin et al., 2021; Zheng et al., 2023): $q(\mathbf{x}_{t-1}|\mathbf{x}_0, \mathbf{x}_t) = q(\mathbf{x}_t|\mathbf{x}_{t-1})q(\mathbf{x}_{t-1}|\mathbf{x}_0)/q(\mathbf{x}_t|\mathbf{x}_0)$. At step $t$, a prediction of $\mathbf{x}_0$ can be decoded given $f_\theta(\mathbf{x}_t, t)$ where $f_\theta$ is the neural network learned during the training. Then the conditional probability $q(\mathbf{x}_{t-1}|\mathbf{x}_0, \mathbf{x}_t)$ can be approximated by $p_\theta(\mathbf{x}_{t-1}|\widehat{\mathbf{x}}_0, \mathbf{x}_t)$ by plugging in $\widehat{\mathbf{x}}_0$, an estimation of $\mathbf{x}_0$. Finally $\mathbf{x}_{t-1}$ will be generated based on $p_\theta(\mathbf{x}_{t-1}|\widehat{\mathbf{x}}_0, \mathbf{x}_t)$. The reverse process for the discrete Markov process is stochastic and requires the neural network's call at every step.

## 2.2 DE-RANDOMIZED PROCESS AND CORRESPONDING REVERSE SAMPLING

In this subsection, we will introduce a non-Markov process such that the joint distribution of $(\mathbf{x}_0, \mathbf{x}_t)$ remains the same as the one defined with Markov process in Section 2.1. However, the induced reverse sampling method is de-randomized and fast.

The new process aims to gradually transform input data $\mathbf{q}_{\text{data}}$ to the noise distribution $\mathbf{q}_{\text{noise}}$ through $T$ intermediate latent variables $\mathbf{x}_1, \dots \mathbf{x}_T$ with the following process:

$$\mathbf{x}_t = b_t \mathbf{x}_{t-1} + (1 - b_t)\mathbf{w}, \tag{4}$$

where $b_t$ is independently drawn from the Bernoulli distribution $\text{Bernoulli}(\beta_t)$ and $\mathbf{w}$ is drawn from the noise distribution $\mathbf{q}_{\text{noise}}$. The only difference between (4) and (1) is that $\mathbf{w}$ is time-invariant during the diffusion. Process in (4) is non-Markov since $q(\mathbf{x}_t|\mathbf{x}_{t-1}, \dots, \mathbf{x}_0)$ doesn't necessarily equals $q(\mathbf{x}_t|\mathbf{x}_{t-1})$.

Although the diffusion process is converted from Markov to non-Markov, the following theorem shows that the condition distribution $q(\mathbf{x}_t|\mathbf{x}_0)$ remains unchanged.

**Theorem 2.1.** *For the non-Markov process in* (4)*, we still have*

$$q(\mathbf{x}_t|\mathbf{x}_0) = \text{Cat}\big(\mathbf{x}_t; \alpha_t \mathbf{x}_0 + (1 - \alpha_t)\mathbf{q}_{\text{noise}}\big),$$

*where $\alpha_t := \Pi_{i=1}^{s} \beta_s$ is specified to decrease from $1$ to $0$ with respect to $t$.*

Using the Bayes' rule, we have $q(\mathbf{x}_t, \mathbf{x}_0) = q(\mathbf{x}_t|\mathbf{x}_0)q(\mathbf{x}_0)$. Thus, according to Theorem 2.1, the joint distribution of $(\mathbf{x}_t, \mathbf{x}_0)$ remains consistent with the process defined in (1). Furthermore, the equation $q(\mathbf{x}_t) = \sum_{\mathbf{x}_0} q(\mathbf{x}_t|\mathbf{x}_0)q(\mathbf{x}_0)$ indicates that probability flow of $q(\mathbf{x}_t)$ with respect to $t$ is preserved.

Therefore, any neural network $f_\theta(\cdot, t)$ learned by the Markov process in (1) can still be used for our non-Markov process (4). Thanks to the de-randomized process, we can now give a simple characterization of $\mathbf{x}_t$'s distribution by introducing the transition time.

**Definition 2.2.** The transition time $\tau$ is the time that the token $\mathbf{x}_t$ transition from $\mathbf{x}_0$ to noise, i.e., $\tau := \min_t \{t|b_t = 0\}$.

*Remark* 2.3. A similar concept is introduced in Hoogeboom et al. (2021a) for the discussion purpose. However, Hoogeboom et al. (2021a) restricts the transition time to be the first time of entering the absorbing state, which is only applicable to absorbing diffusion. Our definition is more general and applies to discrete diffusion with various noise $q_{\text{noise}}$ including multinomial diffusion.

We can get the following process given the transition time $\tau$.

$$\mathbf{x}_t = \mathbb{1}(\tau > t)\mathbf{x}_0 + \mathbb{1}(\tau \leq t)\mathbf{w}, \tag{5}$$

which shows that the token will be a real token $\mathbf{x}_0$ before the time $\tau$ and will be the noise $\mathbf{w}$ after the transition time. Since token only get changed at the transition time $\tau$, we can derive a reverse process based on (5),

$$\mathbf{x}_{t-1} = \mathbb{1}(\tau = t)\mathbf{x}_0 + \mathbb{1}(\tau \neq t)\mathbf{x}_t. \tag{6}$$

Therefore, the process in (6) is de-randomized given transition time $\tau$: as long as we observe $\mathbf{x}_0$ and $\mathbf{x}_t$ for some $t$, then $\mathbf{x}_{t-1}$ becomes known and fixed. It is also worth noting that given $\mathbf{x}_0$ and $\tau$, the exact reverse process (6) is Markovian, since $\mathbf{x}_{t-1}$ solely depends on $\mathbf{x}_0, \tau, \mathbf{x}_t$. Assuming that the learned reverse process is also Markovian given $\mathbf{x}_0$ and $\tau$, then the estimation error of the learned reverse process can be characterized by the Kullback–Leibler (KL) divergence between the reverse process $q$ and the learned reverse process $p_{\boldsymbol{\theta}}$, which can be further decomposed into the summation of the KL divergence at each time $t$. See Appendix B.3 for details.

*Remark* 2.4. Equations (5) and (6) suggest that even though there are $T$ distinct time steps, not every time in the range $1:T$ is crucial for capturing the process. Our primary focus should be on the most significant time step, i.e., the transition time $\tau$. This insight drives our motivation to devise a faster reverse sampling method. Notably, the transition time $\tau$ exhibits randomness and can differ across runs. Hence, the $T$ time steps are not redundant but have significance.

*Remark* 2.5. Song et al. (2020a) introduced the denoising diffusion implicit model (DDIM) for the continuous Gaussian process, as represented in Equation (7) in their paper. Their model is non-Markov and also de-randomized. It can be mathematically expressed as: $\widehat{\mathbf{x}}_{t-1} = \left( \sqrt{\widehat{\alpha}_{t-1}} - \sqrt{\widehat{\alpha}_t(1 - \widehat{\alpha}_{t-1})/(1 - \widehat{\alpha}_t)} \right)\widehat{\mathbf{x}}_0 + \sqrt{(1 - \widehat{\alpha}_{t-1})/(1 - \widehat{\alpha}_t)}\widehat{\mathbf{x}}_t$, where $\widehat{\mathbf{x}}_t$ denotes a continuous diffusion process, and $\widehat{\alpha}$ is its associated coefficient. In this context, our processes as described in Equations (5) and (6) can be considered as discrete counterparts for DDIM. It is worth noting that while Song et al. (2020a) also proposed a model for discrete diffusion in Equation (18) of their paper, their discrete process is randomized and lacks empirical validation. In contrast, our method offers a full de-randomization by utilizing the transition time argument. As we will elaborate in Section 3, our discrete non-Markov diffusion model (DNDM) can achieve a faster sampling speed under the same number of sampling steps. Such efficiency has not been reported in DDIM and stands as a distinctive feature of our discrete diffusion approach.

## 3 ACCELERATED REVERSE SAMPLING

This section will demonstrate that sampling from DNDM can lead to accelerated reverse sampling. Although our algorithm is quite general, we will focus on text generation in the presentation.

In Section 2, we only consider the case of a single token $\mathbf{x} \in \mathbb{R}^K$ being one hot encoding of $K$ categories. In real applications, we are interested in generating a sentence with multiple tokens. So, we extend the terminology in Section 2, and we denote the sequence of tokens at $t$-th time step to be $\mathbf{x}_{t,1:N} = [\mathbf{x}_{t,1}, \ldots, \mathbf{x}_{t,N}]$ where $\mathbf{x}_{t,n}$ is the $n$-th token and $N$ is the sequence length. The noise will be added to each token in a sequence independently. Therefore, each token will have its own transition time defined in Definition 2.2. We denote the transition time for each token $\mathbf{x}_n$ to be $\tau_n$ and further denote the transition time set $\mathcal{T} := \{\tau_n\}_{n=1}^N$. Given the transition times $\tau_n \in \mathcal{T}$, our DNDM can now be extended to the sequence with multiple tokens

$$\mathbf{x}_{t-1,n} = \mathbb{1}(\tau_n = t)\mathbf{x}_{0,n} + \mathbb{1}(\tau_n \neq t)\mathbf{x}_{t,n}, \forall n \in [N]. \tag{7}$$

**Learning the Reverse Process.** We first generate the transition times $\tau_n$ for $n \in [N]$, then we follow (7) to generate the learned reverse process. Since $\mathbf{x}_{0,n}$ is unknown in the process, we will use the neural network evaluation $f_\theta(\cdot, t)$ obtained in Section 2 to predict $\mathbf{x}_{0,n}$. It is worth noting that, given $\mathbf{x}_0$ and $\tau$, the learned reverse process (6) is indeed Markovian, since $\mathbf{x}_{t-1}$ solely depends on $\tau, \mathbf{x}_t$.

In detail, the noisy sequence $\mathbf{x}_{t,1:N}$ is fed into $f_\theta(\cdot, t)$ and collect the output $f_\theta(\mathbf{x}_{t,1:N}, t)$. The tokens generally uses a transformer structure (Hoogeboom et al., 2021b; Austin et al., 2021) and thus $f_\theta(\mathbf{x}_{t,1:N}, t)$ also have $n$-positions $f = [f_1, \ldots, f_N]$. Here $f_n(\mathbf{x}_{t,1:N}, t) \in \mathbb{R}^K$ is a probability vector. We will collect the output $f_n(\mathbf{x}_{t,1:N}, t)$ for $n$-th token. A prediction token $\widehat{\mathbf{x}}_{0,n}$ will then be decoded from $f_n(\mathbf{x}_{t,1:N}, t)$ using different decoding strategies including greedy decoding, top-k decoding and temperature decoding. Note that the decoding strategy here pertains to the approach for obtaining $\widehat{x}_{0,n}$ in traditional NLP tasks and is independent of our diffusion model.

**Transition time.** Transition time, denoted as $\tau$, is crucial in our reverse process. This is because the reverse sampling becomes deterministic upon using (7). Each instance of transition time $\tau$ is a random variable within the set $\{1, 2, \ldots, T\}$. Let's assume it follows the distribution $\mathcal{D}_\tau$. Given the schedule $\{\alpha_t\}_{t=0}^{T}$, we can derive the distribution for $\mathcal{D}_\tau$.

**Theorem 3.1.** *Each specific transition time $\tau_n$ in Definition 2.2 is independent. Furthermore, they collectively adhere to the distribution $\mathcal{D}_\tau$, which obeys the rule $\mathbb{P}(\tau_n = t) = \alpha_{t-1} - \alpha_t$.*

From Theorem 3.1, we discern that the nature of the diffusion model scheduler, $\alpha_t$, clarifies the distribution of $\tau$. Take the linear schedule as an example, as given by (Austin et al., 2021), the relationship is $\alpha_t = 1 - t/T$. This translates to $\mathbb{P}(\tau_n = t) = 1/T$ for every $t$ in the range 1 to $T$. As a result, transition time distributes uniformly across each moment in the set $\{1, \ldots, T\}$. Generally, if we express $\alpha_t$ as $g(t/T)$, then we can simplify to $\mathbb{P}(\tau_n = t) = g((t - 1)/T) - g(t/T)$, which further refines to $(1/T)|g'(t/T)| + o(1/T)$. This indicates that transitions are more likely where $|g'|$ is large.

In practical applications, we have observed that the shape of the transition time does not need to match the theoretical prediction schedule exactly. A more efficient way is to give an approximate schedule with a Beta distribution. Essentially, we first sample a time $t \in [0, 1]$ from a Beta distribution, then adjust these samples to fit by multiplying $T$ and round them to acquire the integer.

**De-randomized Accelerated Sampling.** According to (7), a token $\mathbf{x}_{t-1,n}$ is updated only if step $t$ is the transition time for the $n$-th token. If step $t$ is not the transition time for any token, the sentence from the previous step can be directly copied: $\mathbf{x}_{t-1,1:N} = \mathbf{x}_{t,1:N}$. As a result, there is no need to evaluate $f_n(\mathbf{x}_{t,1:N}, t)$ for the current step. Our attention, therefore, can be solely centered on the transition set $\mathcal{T}$, necessitating function evaluations only for $t$ within $\mathcal{T}$. Given that the cardinality of $\mathcal{T}$ is at most $|\mathcal{T}| = \min\{N, T\}$, the function evaluations decrease from $T$ to $|\mathcal{T}|$ for a particular example. Further insights into the sampling based on this approach are detailed in Algorithm 1. For our method, when $N$ is fixed while $T \to \infty$, the total NFE will reach $N$. On the other hand, when $T$ is fixed and $N \to \infty$, the NFE will reach $T$. Such observation can be rigorously inferred from our Theorem D.1. Therefore, our framework bridges two extremes of (fixed N, infinite T) and (infinite N, finite T). It is worth noting that the auto-regressive diffusion model (ARDM) Hoogeboom et al. (2021a) can also achieve at most $N$ NFE when $T = \infty$. ARDM is primarily focused on infinite timesteps, while our method here accelerates sampling for finite time steps. More detailed discussion and theoretical analysis (Theorem D.1) can be found in Section D. Additional experiment in §D also demonstrates that our DNDM achieves an NFE that is less than half of the original sampling method. Another property of Algorithm 1 is that it does not require intricate probability computations, further speeding up the sampling process. By incorporating the forward process with different noises, we can develop DNDM-Multi and DNDM-Absorb, which accelerate the Multinomial and Absorbing sampling methods respectively. Recent works have demonstrated that the quality of samples can be enhanced by utilizing supplementary information derived from the neural network, $f_\theta(\cdot, t)$ (Ghazvininejad et al., 2019; Savinov et al., 2021; Chang et al., 2022; He et al., 2022; Zheng et al., 2023). Our DNDM can also be improved using this idea. We call it a discrete non-Markov Diffusion Model with Top-k Transition Time (DNDM-$k$). Due to the limit of the pages, we leave the detailed Algorithm and discussion to Appendix E.

## 4 CONTINOUS-TIME (INFINITE STEP) REVERSE SAMPLING

In the context of continuous state spaces, continuous-time processes have been proposed to accommodate algorithms that offer faster sampling speeds and enhanced sample quality (Jolicoeur-Martineau et al., 2021; Zhang & Chen, 2022; Salimans & Ho, 2022; Chung et al., 2022; Song et al., 2020b; Dockhorn et al., 2021). However, the utilization of continuous-time schemes to discrete-state spaces remains underexplored. Furthermore, later in Section 5.1, we will compare the performance of sampling with 50 steps to sampling with 1000 steps. It's important to note that choosing a finer step size in reverse sampling can result in higher BLEU scores, as demonstrated in Tables 1 and 2. This highlights the need for the development of a more flexible time sampling schedule.

Campbell et al. (2022) first designed a continuous framework for discrete-time diffusion for the Markovian process and randomized sampling, however not in our non-Markovian setting. In this section, we present the continuous-time extension of our non-Markovian process and DNDM algorithm. This approach not only provides flexibility by allowing arbitrary time points but also demonstrates improved BLEU scores on some tasks.

---

**Algorithm 1** Sampling From DNDM

**Require:** Trained prediction function $f_\theta(\cdot, t)$, $\mathbf{q}_{\text{noise}}$, $\mathcal{D}_\tau$
1: **for** $n = 1 \ldots N$ **do**
2:     Initiate each token $\mathbf{x}_{T,n} \sim \mathbf{q}_{\text{noise}}$
3:     Initiate the transition time $\tau_n \sim \mathcal{D}_\tau$
4: **end for**
5: Collect transition time set $\mathcal{T} = \{\tau_n\}_{n=1}^N$
6: **for** $t = T \ldots 1$ **do**
7:     **if** $t \in \mathcal{T}$ **then**
8:         Generate $\widetilde{\mathbf{x}}_{0,1:N}$ from $f_\theta(\mathbf{x}_{t,1:N}, t)$
9:         **for** $n = 1 \ldots N$ **do**
10:            Update $\mathbf{x}_{t-1,n}$ based on condition of $\tau_n$
11:         **end for**
12:     **else**
13:         Update $\mathbf{x}_{t-1,1:N} = \mathbf{x}_{t,1:N}$
14:     **end if**
15: **end for**
16: **return** $\mathbf{x}_{0,1:N}$

**Algorithm 2** Sampling from DNDM-C

**Require:** Trained prediction function $f_\theta(\cdot, t)$, $\mathbf{q}_{\text{noise}}$, $\mathcal{D}_\tau$
1: **for** $n = 1 \ldots N$ **do**
2:     Initiate each token $\mathbf{x}_{T,n} \sim \mathbf{q}_{\text{noise}}$
3:     Initiate the transition time $\tau_n \sim \mathcal{D}_\tau$
4: **end for**
5: Collect and arrange transition times in $\mathcal{T}$
6: **for** $k = 1 \ldots N$ **do**
7:     Generate $\widetilde{\mathbf{x}}_{0,1:N}$ using $f_\theta(\mathbf{x}_{t_{k-1},1:N}, t_k)$
8:     **for** $n = 1 \ldots N$ **do**
9:         Update $\mathbf{x}_{t_{k+1},n}$ based on condition of $\tau_n$
10:     **end for**
11: **end for**
12: **return** $\mathbf{x}_{0,1:N}$

---

**Continuous-time forward and backward process.** We recall that the forward process described in (4) can be sampled from $\mathbf{x}_{0,n}$ through the following process:

$$\mathbf{x}_{t,n} = \alpha_t \mathbf{x}_{0,n} + (1 - \alpha_t)\mathbf{q}_{\text{noise}}, \quad \alpha_t = \prod_{i=1}^{s} \beta_s. \tag{8}$$

In the previous section, we are constrained to discrete time steps, where we must define a maximum step, denoted as $T$. The values of $\mathbf{x}_t$ are computed only for $t = 1, \ldots, T$. As a result, during the training process, it is only possible to predict the $\mathbf{x}_0$ at these predetermined time steps. This constraint confines the computation of our reverse process exclusively to these fixed time stamps.

To derive the continuous limit of (8), for each $T$ we rescale (8) to a diffusion process on $[0, 1]$, e.g. $\mathbf{x}_{T,n} = \widehat{\mathbf{x}}_{1,n}, \mathbf{x}_{0,n} = \widehat{\mathbf{x}}_{0,n}$, and $\mathbf{x}_{t,n} = \widehat{\mathbf{x}}_{t/T,n}$. Therefore, when $T \to \infty$, $\widehat{\mathbf{x}}_{t,n}$ represents the continuous process that has value at arbitrary $t \in [0, 1]$. If the choice of $\alpha_t$ for each $T$ is scale-invariant, we can define a continuous function $\alpha(t)$ as the continuous $\alpha$ schedule of the discrete counterpart[2]. More specifically, we obtain

$$\widehat{\mathbf{x}}_{t,n} = \alpha(t)\widehat{\mathbf{x}}_{0,n} + (1 - \alpha(t))\mathbf{q}_{\text{noise}}, \quad t \in [0, 1]. \tag{9}$$

For the reverse-time process, we define the transition time set $\mathcal{T} := \{\tau_n\}_{n=1}^N$ consistent with Theorem 3.1 and sample it from $\mathbb{P}(\tau_n = t) = -\alpha'(t)$. With $\mathcal{T}$ defined, the updates to $\mathbf{x}_{t,n}$ only occur at $\{\tau_n\}$. Consequently, we arrange $\tau_n$ to obtain an ordered sequence $\tau_{n_k}$, where $\tau_{n_1} < \tau_{n_2} < \ldots < \tau_{n_N}$. When omitting the infinitely many time steps between $\tau_{n_k}$ and $\tau_{n_{k-1}}$. The resulting reverse process is then given by:

$$\mathbf{x}_{\tau_{n_{k-1}},n} = \mathbb{1}(\tau_n = \tau_{n_{k-1}})\mathbf{x}_{0,n} + \mathbb{1}(\tau_n \neq \tau_{n_{k-1}})\mathbf{x}_{\tau_{n_k},n}, \quad \forall n \in [N]. \tag{10}$$

This limit is taken with infinitesimally small time steps, yielding an exact solution rather than an approximate transition time schedule when considering the partition of time steps. The resulting algorithm is shown in Algorithm 2.

**Comparison with ARDM (Hoogeboom et al., 2021a).** Autoregressive Diffusion Model (ARDM) Hoogeboom et al. (2021a) is a discrete diffusion model built upon the autoregressive nature of data. ARDM is shown to be equivalent to a continuous-time absorbing diffusion model and thus provides a unique perspective for discrete diffusion. For continuous-time ($T = \infty$) reverse sampling, both ARDM and our method achieve $N$ NFEs. Compared with ARDM, our method provides a unified framework including both absorbing and multinomial diffusions, applicable to both finite time and continuous time diffusions. For infinite timesteps, Hoogeboom et al. (2021a) also proposed an advanced parallelizing technique that can reduce NFE according to the log-likelihood, which we have not considered in DNDM-C.

---

[2]If we represent $\alpha_t$ with maximum step $T$ as $\alpha_t(T)$, the scale-invariant property states that $\alpha_{ct}(cT) = \alpha_t(T)$. The simplest example of such an $\alpha_t$ schedule is $\alpha_t(T) = 1 - t/T$, under which $\alpha(t) = 1 - t$.

## 5 EXPERIMENTS

In this section, we evaluate DNDM and demonstrate its superior performance on two types of tasks: conditional sequence-to-sequence text generation (i.e., machine translation) and unconditional text generation. For the fairness of comparison, all the experiments are conducted using a single NVIDIA RTX A6000 GPU with 48 GB memory. Additional experiment details are provided in Appendix F.

### 5.1 CONDITIONAL TEXT GENERATION

We first evaluate the ability of DNDM on conditional text generation by machine translation tasks. Following (Zheng et al., 2023), we process the raw text with the Byte Pair Encoder (BPE) (Sennrich et al., 2016) to construct the vocabulary, which consists of the words and subwords of both the source and the target languages. We conduct our experiments using the FairSeq toolkit (Ott et al., 2019), which employs a model consisting of an encoder and a decoder. The encoder takes the source text as input, while the decoder generates the target text.

**Datasets.** We use the following three datasets to compare with the baselines for machine translation tasks: (1) `IWSLT14 DE-EN` (Cettolo et al., 2014), a dataset with German as the source language and English as the target language. It consists of $173972$ examples (sentence pairs), and each of the validation set and the testing set accounts for around $4.2\%$ of the dataset; (2) `WMT14 EN-DE` (Bojar et al., 2014), which is an English-to-German translation dataset consisting of $3967182$ examples. Each of the validation set and the testing set accounts for around $0.076\%$ of the dataset; and (3) `WMT16 EN-RO` (Bojar et al., 2016), which is an English-to-Russian translation dataset consisting of $612317$ examples. Each of the validation sets and the testing set accounts for around $0.33\%$ of the dataset. For all machine translation datasets, the train-validation-test split is fixed across all experiments to ensure fair comparison.

**Performance metrics.** We use the BLEU score (Papineni et al., 2002) to evaluate the machine translation quality, where the BLEU score is calculated based on the similarity between the actual target sequence and the predicted target sequence. The sampling speed is measured by wall-clock time (in second).

**Baselines.** The main baselines we are comparing with are RDM and RDM-$k$ from Zheng et al. (2023). Here, we use RDM and RDM-$k$ to denote the sampling method proposed in their paper with and without the usage of top-$k$ selection for the token generation technique (see Appendix E for more details), respectively. RDM and RDM-$k$ are applied to two previously proposed state-of-the-art discrete diffusion models: Multinomial Diffusion (Hoogeboom et al., 2021b) and D3PM (Austin et al., 2021), where the latter is a typical absorbing discrete diffusion model, so in the following sections the term "absorbing diffusion" refers to it.

**Diffusion type.** There are two types of diffusion proposed for discrete diffusion models: (1) multinomial diffusion (Hoogeboom et al., 2021b), and (2) absorbing diffusion (Austin et al., 2021).

**Decoding type.** In our experiments, we found that argmax decoding always outperforms normal decoding across all the tasks. Therefore, we only report the experiment results based on argmax decoding.

**Results and Discussion.** Tables 1 and 2 present the performance evaluations of our algorithms in machine translation tasks. Table 1 presents results for multinomial diffusion, while Table 2 displays results for absorbing diffusion. Our reported time and BLEU scores are averaged over 5 repeated experiments, except for the baseline RDM experiment[3].

From Tables 1 and 2, we observe that methods based on DNDM significantly accelerate the sampling process compared to baseline diffusion models. This acceleration allows for greater flexibility in increasing the number of steps (up to infinity) without imposing a significant computational burden. Consequently, our sampling method demonstrates an improved overall BLEU score. More specifically, as we can see in each column of Tables 1 and 2, more sampling steps lead to better generation quality (BLEU) at the expense of longer sampling time. For RDM-based methods, generation time increases linearly with the number of sampling steps. On the contrary, for our DNDM-based method, generation time only increases marginally (See Figure 4 in Section G). As a result of the difference in the growing speed of sampling time with respect to sampling steps, the more sampling steps, the more speedup DNDM can obtain. Notably, when the sampling steps increase to 1000, DNDM can accelerate the

---

[3]Due to computational intensity, we did not repeat the 1000-step sampling for the RDM baseline. However, reproducing it was deemed unnecessary as the sampling time is largely stable across repeated experiments, and the precise averaged timing is not critical for demonstrating the speed improvement of DNDM.

sampling process 30-60 times compared to RDM. This remarkable acceleration is attributed to the introduction of transition times and the elimination of unnecessary steps, as discussed in Section 3. Continuous-time results, as the ultimate limit of increasing sampling steps, are presented in the last row of each dataset with the tag $\infty$. Given that the results with 1000 steps consistently outperform those with 50 steps, we compare $\infty$ with 1000 steps in Table 1 and 2. For IWSLT14 and WMT16, where the generation BLEU score is relatively high, we observe a consistent performance improvement of up to 0.3 in BLEU score when utilizing the DNDM-C algorithm, with the exception of a single case in the absorbing diffusion setting for WMT16 without the use of top-$k$ selection. The performance gain of the continuous-time method on WMT14 is less significant, with both drops and gains. However, WMT14 itself has not reached a high level of performance, with a BLEU score significantly lower than other datasets. In general, training WMT14 poses challenges across all diffusion models, including multinomial diffusion (Hoogeboom et al., 2021b), absorbing diffusion (Austin et al., 2021), and RDM diffusion (Zheng et al., 2023), etc. We defer a more detailed discussion on WMT14 to Appendix F.1. Finally, when compared with the results obtained with 50 steps, the performance of DNDM-C demonstrates improvement consistently. Furthermore, we note that regardless of the dataset or the method (i.e., RDM or DNDM) employed, top-$k$ token generation consistently outperforms vanilla methods. This approach enhances the BLEU score by approximately 1-2 points without introducing significant increases in sampling time.

| Dataset | Steps | RDM-Multi | | DNDM-Multi | | RDM-$k$-Multi | | DNDM-$k$-Multi | |
|---------|-------|-----------|-----------|-----------|-----------|-----------|-----------|-----------|-----------|
| | | BLEU | Time (s) | BLEU | Time (s) | BLEU | Time(s) | BLEU | Time (s) |
| IWSLT14 | 25 | **31.26** | 166.9 | 30.95 | **52.9** | **32.82** | 161.9 | 32.30 | **52.6** |
| | 50 | **31.50** | 328.6 | 31.45 | **83.9** | **32.82** | 321.2 | 32.80 | **93.2** |
| | 1000 | 31.69 | 6308.9 | **31.82** | **191.3** | 32.64 | 6321.3 | **33.15** | **191.5** |
| | $\infty$ | - | - | **31.89** | **225.2** | - | - | **33.44** | **228.1** |
| WMT14 | 25 | **25.25** | 237.3 | 25.01 | **90.7** | **26.03** | 230.9 | 25.98 | **90.5** |
| | 50 | **25.75** | 466.1 | 25.33 | **138.4** | 26.14 | 500.2 | **26.37** | **138.3** |
| | 1000 | 25.66 | 8996.7 | **25.71** | **265.4** | 25.82 | 8991.7 | **26.88** | **265.5** |
| | $\infty$ | - | - | **24.79** | **307.5** | - | - | **26.39** | **307.3** |
| WMT16 | 25 | **32.29** | 145.2 | 31.97 | **36.4** | **33.12** | 143.5 | 32.94 | **36.4** |
| | 50 | **32.53** | 286.1 | 32.50 | **63.2** | **33.41** | 312.4 | 33.26 | **62.7** |
| | 1000 | 32.63 | 5588.9 | **32.86** | **171.4** | 33.67 | 5601.0 | **33.79** | **171.2** |
| | $\infty$ | - | - | **32.91** | **196.4** | - | - | **33.86** | **196.3** |

Table 1: BLEU score comparison of multinomial diffusion on machine translation benchmarks IWSLT14 DE-EN, WMT14 EN-DE, and WMT16 EN-RO. The blue background highlights our algorithms, and the bold number indicates the best performance within each row and each setting (i.e., with or without top-k).

| Dataset | Steps | RDM-Absorb | | DNDM-Absorb | | RDM-$k$-Absorb | | DNDM-$k$-Absorb | |
|---------|-------|-----------|-----------|-----------|-----------|-----------|-----------|-----------|-----------|
| | | BLEU | Time (s) | BLEU | Time (s) | BLEU | Time(s) | BLEU | Time (s) |
| IWSLT14 | 25 | 31.58 | 116.3 | **32.43** | **67.2** | **34.50** | 108.9 | 34.14 | **67.3** |
| | 50 | 31.80 | 227.2 | **32.63** | **95.9** | **34.58** | 213.9 | 34.34 | **96.2** |
| | 1000 | 31.91 | 4197.4 | **32.93** | **161.1** | **34.60** | 4205.9 | 34.56 | **162.3** |
| | $\infty$ | - | - | **33.03** | **174.6** | - | - | **34.65** | **180.7** |
| WMT14 | 25 | 24.97 | 116.4 | **25.79** | **68.1** | **27.50** | 107.5 | 27.18 | **68.0** |
| | 50 | 24.95 | 231.1 | **26.10** | **102.0** | **27.73** | 255.2 | 27.66 | **102.5** |
| | 1000 | 25.22 | 4169.4 | **26.43** | **178.3** | 27.75 | 4167.4 | **27.82** | **179.1** |
| | $\infty$ | - | - | **26.50** | **180.1** | - | - | **27.50** | **181.2** |
| WMT16 | 25 | 32.86 | 75.5 | **33.20** | **41.2** | 33.92 | 69.9 | **33.96** | **41.4** |
| | 50 | 32.93 | 148.4 | **33.30** | **62.5** | 34.10 | 166.1 | **34.20** | **62.7** |
| | 1000 | 33.25 | 2951.7 | **33.60** | **121.3** | **34.44** | 2718.7 | 34.38 | **122.7** |
| | $\infty$ | - | - | **33.42** | **121.8** | - | - | **34.41** | **121.9** |

Table 2: BLEU score comparison of absorbing diffusion on machine translation benchmarks IWSLT14 DE-EN, WMT14 EN-DE, and WMT16 EN-RO. The blue background highlights our algorithms, and the bold number indicates the best performance within each row and each setting (i.e., with or without top-k).

**Scaling law in sampling speed.** For illustrative purposes, we use the example of `IWSLT14` to visualize how the sample quality scales regarding sampling speed for different methods. In Figure 1, we observe the trend of the BLEU score in relation to computational time. Each line in the legend represents a different sampling algorithm, and a steeper slope indicates a larger marginal gain when sampling for longer periods. Figure 1 demonstrates that our algorithm displays nearly linear growth in BLEU score over the log of time, which is remarkable in contrast with the flat curve of the baseline. Particularly, for multinomial diffusion, the BLEU score increases by 1 in less than 60 seconds of additional sampling time. For absorbing diffusion, DNDM outperforms RDM before RDM samples 50 steps. In Tables 5 and 6 in Appendix §D, we further use the average number of function evaluations (NFE) to measure the improved efficiency within the specified number of sampling steps. Additionally, in Figure 2, we visualize how the BLEU score and the generated text change throughout the sampling process.

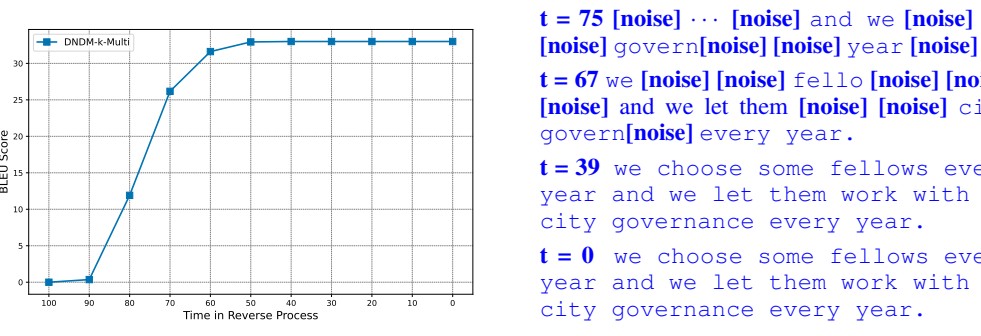

|  (a) Multinomial Diffusion | (b) Absorbing Diffusion |
|---|---|

Figure 1: Generation quality to generation time comparison on `IWSLT14`. $x$-axis: computational time in seconds; $y$-axis: BLEU score.

**t = 100**  **[noise] [noise] [noise] [noise]** $\cdots$

**t = 75 [noise]** $\cdots$ **[noise]** and we **[noise]** $\cdots$ **[noise]** govern**[noise] [noise]** year **[noise]**

**t = 67** we **[noise] [noise]** fello **[noise] [noise] [noise]** and we let them **[noise] [noise]** city govern**[noise]** every year.

**t = 39** we choose some fellows every year and we let them work with city governance every year.

**t = 0** we choose some fellows every year and we let them work with city governance every year.

(a) The BLEU Score in the Generation Process  (b) Text in the Generation Process

Figure 2: We demonstrate the 100-step generation process of DNDM-$k$-Multi as an example, where the left is the change of the BLEU score along the generation process, and the right is the text at different time steps. As the time goes from 100 to 0, noise is gradually removed until the corresponding English text emerges. Since the transition time follows a Beta distribution as described in Section 3, the majority of transitions occur near the starting time.

## 5.2 UNCONDITIONAL TEXT GENERATION

In unconditional text generation, we focus on language modeling tasks, where the goal is to generate language data similar to the provided training dataset. In this task, no input text is given during sampling, and the neural network directly learns $q(\mathbf{x}_0|\mathbf{x}_t)$.

**Datasets.** The natural language generation task is evaluated on two language datasets following Hoogeboom et al. (2021b): `text8` and `enwik8`. Both datasets are from Wikipedia, but their contents are highly distinct. In `text8`, the plain text consists of English words (all the letters are in lower case) and spaces, and it is tokenized into 26 characters and one blank space, resulting in 27 categories. In contrast to the cleanness of `text8`, `enwik8` preserves the original XML dump contents, and there exist various special symbols in its raw text, so its text is tokenized into 1 Byte, resulting in 256 categories. We utilize `text8` dataset with sequence length 256 and `enwik8` dataset with sequence length 320. The train/val/test splits are 9e7/5e6/5e5 for both `text8` and `enwik8`.

**Performance metrics.** Our evaluation of text generation quality relies on the perplexity score. When generating `text8` data, we calculate perplexity scores using the GPT2 model, while for enwik8 data generation, we employ the GPT2-large model. The sampling speed is measured in seconds.

**Baselines.** We compare our proposed DNDM on unconditional text generation task with the vanilla Multinomial Diffusion (Hoogeboom et al., 2021b).

**Results and discussion.** Table 3 displays the performance of our algorithms in text generation tasks. We run the multinomial diffusion model on the `text8` dataset for 1000 diffusion steps and on the `enwik8` dataset for 4000 diffusion steps. Our DNDM-based algorithms outperform the vanilla sampling algorithm used in Hoogeboom et al. (2021b) in terms of both sampling time and perplexity score. Specifically, for the `text8` dataset, DNDM-based algorithms are 5 times faster than the vanilla algorithm. For the `enwik8` dataset, DNDM-based algorithms are 14 times faster than the vanilla algorithm.

| | | Vanilla | DNDM |
|---|---|---|---|
| `text8` | Perplexity | 1,465.75 | **600.02** |
| | Time (s) | 135.9 | **31.1** |
| `enwik8` | Perplexity | 801.78 | **556.78** |
| | Time (s) | 602.8 | **47.4** |

Table 3: Comparison of different sampling methods for unconditional text generation (multinomial diffusion) on `text8` and `enwik8` benchmarks. Sampling time is computed by generating a single text sample of length 256 for `text8` and length 320 for `enwik8`, averaged over 10 runs. The blue background represents our algorithms, and the bold number indicates the optimal value.

## 6 CONCLUSION AND FUTURE WORK

This paper presents a novel discrete non-Markov diffusion model (DNDM) accompanied by an accelerated algorithm designed to boost the sampling speed in a discrete-state space. Our discrete diffusion model incorporates "transition time set" latent variables, establishing itself as an efficacious diffusion and data generation method. Thanks to our accelerated technique, we significantly decrease the number of calls to the neural network without sacrificing the quality of samples. We also introduce an infinite-step sampling algorithm, DNDM-C, which exhibits superior sample quality relative to its finite-step counterparts. While this study focuses on text generation using non-autoregressive models, a promising direction for future exploration is applying our method to other tasks like audio and image generation, as well as other architectures such as the GPT model (an auto-regressive model).

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

## A    RELATED WORK

**Continous Diffusion Models.** Generative modeling via continuous-time stochastic process has been investigated thoroughly in a series of work (Movellan, 2008; Lyu, 2012; Sohl-Dickstein et al., 2009; Bengio et al., 2014; Alain et al., 2016; ALIAS PARTH GOYAL et al., 2017; Bordes et al., 2017). The two lines of probabilistic modeling, *denoising diffusion probabilistic model* (Sohl-Dickstein et al., 2015; Ho et al., 2020) and *score matching with Langevin dynamics* (Song & Ermon, 2019) are unified by Song et al. (2020b) through introducing the SDE framework for SGM. Based on it, subsequent works (Dockhorn et al., 2021; Nachmani et al., 2021; Vahdat et al., 2021) introduced a more complex diffusion process to improve the generation efficiency and quality. On the other hand, the score-based sampling process is time-consuming and has attracted much attention for improvements in speed (San-Roman et al., 2021; Watson et al., 2021; Kong & Ping, 2021; Karras et al., 2022; Song et al., 2023). "Gotta go fast" (GGF), an SDE solver with adaptive step size tailored to SGM, is proposed in Jolicoeur-Martineau et al. (2021). Song et al. (2020a) introduced a non-Markov diffusion process that corresponds to a deterministic sampling process, enabling the generation of high-quality samples more rapidly. Dockhorn et al. (2022); Liu et al. (2022) proposed a high-order SDE/ODE solver to achieve lower discretization error. Lu et al. (2022); Zhang & Chen (2022) leveraged the semi-linear structure of reverse ODE to reduce the discretization error and achieve state-of-the-art sampling speed.

**Discrete Diffusion Models.** Research on discrete diffusion models was initiated by Sohl-Dickstein et al. (2015), who investigated diffusion processes over binary random variables. The methodology was expanded upon by Ho et al. (2020), integrating categorical random variables through transition matrices with uniform probabilities. Though Song et al. (2020a) suggested a similar extension in their supplementary content, they abstained from experimenting with this model type. Later on, Austin et al. (2021) unveiled a more intricate framework for diffusion concerning categorical random variables, enhancing the discrete diffusion models by merging them with Masked language models (MLMs). Contemporary research has furthered this domain by introducing features like editing-based operations (Jolicoeur-Martineau et al., 2021; Reid et al., 2022), auto-regressive diffusion models (Hoogeboom et al., 2021a; Ye et al., 2023), the evolution of a continuous-time structure (Campbell et al., 2022), and the exploration of neural network analogs for learning (Sun et al., 2022). Additionally, Zheng et al. (2023) introduced a re-parameterized loss and an associated sampling technique, attaining commendable outcomes in fewer iterations. Our contributions run parallel to these aforementioned studies.

## B    ADDITIONAL DETAILS OF DISCRETE DIFFUSION

In our paper, we treat all the $\mathbf{x}, \mathbf{q}_{\text{noise}}$ as a row vector and treat $\mathbb{1}$ as a column vector with all elements equal 1.

### B.1    DE-RANDOMIZATION

In Section 2, we introduced two different diffusion processes, the Markov process in Equation (1) and the non-Markov process in Equation (4). In this section, we will explain why they are different but will result in the same joint distribution of $(\mathbf{x}_0, \mathbf{x}_t)$ for every time step $t$. Since $\mathbf{q}(\mathbf{x}_0)$ keeps the same, we only need to prove that the conditional distribution $\mathbf{q}(\mathbf{x}_t|\mathbf{x}_0)$ is the same for the two processes.

**Markov Process.** Equation 1 is a Markov process since $\mathbf{w}_n$ is independent with $\mathbf{x}_{t-1}, \dots, \mathbf{x}_0$, so $\mathbf{x}_t$ is independent of all the past states given the present state. This can also be inferred from the following distribution, which does not depend on $\mathbf{x}_0, \dots, \mathbf{x}_{t-2}$,

$$q(\mathbf{x}_t|\mathbf{x}_{t-1}) = \text{Cat}\big(\mathbf{x}_t; \beta_t \mathbf{x}_{t-1} + (1 - \beta_t)\mathbf{q}_{\text{noise}}\big). \tag{11}$$

Denote $\mathbf{Q}_t := \beta_t \mathbf{I} + (1 - \beta_t)\,\mathbb{1}\,\mathbf{q}_{\text{noise}}$, then we have that

$$\mathbf{x}_{t-1}\mathbf{Q}_t = \beta_t \mathbf{x}_{t-1} + (1 - \beta_t)\mathbf{x}_{t-1}\,\mathbb{1}\,\mathbf{q}_{\text{noise}}$$
$$= \beta_t \mathbf{x}_{t-1} + (1 - \beta_t)\mathbf{q}_{\text{noise}},$$

where the last equality holds due to the fact that $\mathbf{x}_{t-1}$ is a one hot vector and thus $\mathbf{x}_{t-1}\mathbb{1} = 1$. Therefore, we can rewrite Equation (11) as $q(\mathbf{x}_t|\mathbf{x}_{t-1}) = \text{Cat}\big(\mathbf{x}_t; \mathbf{x}_{t-1}\mathbf{Q}_t\big)$. Then, it is a Markov process with transition kernel $\mathbf{Q}_t$. So $q(\mathbf{x}_t|\mathbf{x}_0) = \text{Cat}\big(\mathbf{x}_t; \mathbf{x}_0 \mathbf{Q}_0 \dots \mathbf{Q}_t\big)$ (Austin et al., 2021). We can then have that

$$\mathbf{Q}_0 \dots \mathbf{Q}_t = [\beta_0 \mathbf{I} + (1 - \beta_0)\,\mathbb{1}\,\mathbf{q}_{\text{noise}}] \dots [\beta_t \mathbf{I} + (1 - \beta_t)\,\mathbb{1}\,\mathbf{q}_{\text{noise}}]$$
$$= \Pi_{s=0}^{t}\beta_s \mathbf{I} + (1 - \Pi_{s=0}^{t}\beta_s)\,\mathbb{1}\,\mathbf{q}_{\text{noise}},$$

where the last equality holds since identity matrix $\mathbf{I}$ multiplying any vector equals the vector itself and $\mathbb{1}\,\mathbf{q}_{\text{noise}}\,\mathbb{1}\,\mathbf{q}_{\text{noise}} = \mathbb{1}(\mathbf{q}_{\text{noise}}\,\mathbb{1})\mathbf{q}_{\text{noise}} = \mathbb{1}\,\mathbf{q}_{\text{noise}}$. Therefore, we have that

$$q(\mathbf{x}_t|\mathbf{x}_0) = \text{Cat}\big(\mathbf{x}_t; \Pi_{s=0}^{t}\beta_s \mathbf{x}_0 + (1 - \Pi_{s=0}^{t}\beta_s)\mathbf{q}_{\text{noise}}\big) = \text{Cat}\big(\mathbf{x}_t; \alpha_t \mathbf{x}_0 + (1 - \alpha_t)\mathbf{q}_{\text{noise}}\big),$$

where the last equality holds due to the definition $\alpha_t = \Pi_{s=0}^{t}\beta_s$. This gives rise to why the Markov process (1) will result in conditional distribution $q(\mathbf{x}_t|\mathbf{x}_0) = \text{Cat}\big(\mathbf{x}_t; \alpha_t \mathbf{x}_0 + (1 - \alpha_t)\mathbf{q}_{\text{noise}}\big)$.

**Non-Markov Process.** Notice that our DNDM is defined by

$$\mathbf{x}_t = b_t \mathbf{x}_{t-1} + (1 - b_t)\mathbf{w},$$

where $\mathbf{w}$ is fixed for any time $t$. Therefore, $\mathbf{w}$ is no longer independent with $\mathbf{x}_0, \ldots, \mathbf{x}_{t-1}$. Therefore, we can't define the transition kernel and compute $\mathbf{q}(\mathbf{x}_t|\mathbf{x}_0)$ by using the property of Markov. Therefore, we need to advance the technique to calculate the conditional distribution.

*Proof of Theorem 2.1.* By Equation (4), we can derive the following explicit expression for a recursive sequence,

$$\begin{aligned}
\mathbf{x}_t &= b_1 \ldots b_t \mathbf{x}_{0,n} + \sum_{s=1}^{t}(1 - b_s)b_{s+1} \ldots b_t \mathbf{w} \\
&= b_1 \ldots b_t \mathbf{x}_0 + (1 - b_1 \ldots b_t)\mathbf{w} \\
&= a_t \mathbf{x}_0 + (1 - a_t)\mathbf{w},
\end{aligned}$$

where second equality is by cancellation of terms, the last inequality holds by defining $a_t = b_1 \ldots b_t$. Since $a_t$ either equals to 1 or 0. Besides, $a_t$ equals 1 if and only if $b_1 = b_2 = \ldots = b_t = 1$, so we have that $a_t$ follows Bernoulli distribution $\text{Bernoulli}(\beta_1 \ldots \beta_t) = \text{Bernoulli}(\alpha_t)$ where $\alpha_t = \Pi_{i=1}^{t}\beta_s$. Therefore, we can conclude that $\mathbf{q}(\mathbf{x}_t|\mathbf{x}_0) = \text{Cat}\big(\mathbf{x}_t; \alpha_t \mathbf{x}_0 + (1 - \alpha_t)\mathbf{q}_{\text{noise}}\big)$, which completes the proof. $\qquad\square$

### B.2 TRAINING OBJECTIVE

In Section 2.1, we have introduced a general form of the training objective,

$$\frac{1}{T}\sum_{t=0}^{T} L_t(f_\theta(\cdot, t), S_t).$$

Next, we will detail the loss $L_t$ for different methods.

Hoogeboom et al. (2021b) utilized $L_t$ derived from the negative variational bound. In detail,

$$L_t\big(f, q(\mathbf{x}_t, \mathbf{x}_0)\big) = \text{KL}\big(\text{Cat}(\mathbf{x}; \boldsymbol{\theta}_{\text{post}}(\mathbf{x}_t, \mathbf{x}_0)\big|\text{Cat}(\boldsymbol{\theta}_{\text{post}}(\mathbf{x}_t, \widehat{\mathbf{x}}_0))\big), \tag{12}$$

where $\widehat{\mathbf{x}}_0 = f_\theta(\mathbf{x}_t, t)$, $\boldsymbol{\theta}_{\text{post}} = (\beta_t \mathbf{x}_t + (1 - \beta_t)/K\,\mathbb{1}^\top) \odot (\alpha_{t-1}\mathbf{x}_0 + (1 - \alpha_{t-1})/K\,\mathbb{1}^\top)$ and $\boldsymbol{\theta}_{\text{post}} = (\beta_t \mathbf{x}_t + (1 - \beta_t)/K\,\mathbb{1}^\top) \odot (\alpha_{t-1}\widehat{\mathbf{x}}_0 + (1 - \alpha_{t-1})/K\,\mathbb{1}^\top)$. This loss evolves KL divergence between two categorical distributions.

Building on this foundation, Austin et al. (2021) introduced an auxiliary denoising objective to strengthen the data predictions $\mathbf{x}_0$ at each time step. In detail, the auxiliary objective is as follows,

$$\mathbb{E}_{q(\mathbf{x}_t, \mathbf{x}_0)}\Big[-\log \widetilde{p}_\theta(\mathbf{x}_0|\mathbf{x}_t)\Big],$$

where $\widetilde{p}_\theta(\mathbf{x}_0|\mathbf{x}_t)$ is a function of $f_\theta(\mathbf{x}_t, t)$ and the auxiliary loss term is minimized exactly when $\widetilde{p}_\theta(\cdot|\mathbf{x}_t)$ has all its mass on the data point $\mathbf{x}_0$.

Furthering the advancements, Zheng et al. (2023) put forth a reparametrized loss $L_t$ that incorporates a re-weighted parameter $\lambda_t$. The detailed loss is

$$L_t\big(f, q(\mathbf{x}_t, \mathbf{x}_0)\big) = \mathbb{E}_{q(\mathbf{x}_t, \mathbf{x}_0)}\Big[-\lambda_{t-1}\mathbf{x}_0 \cdot \log(f_\theta(\mathbf{x}_t, t))\Big].$$

This loss can be related to the standard multi-class cross-entropy loss function, which is also simple and powerful. That's why we consider Zheng et al. (2023) as the baseline model.

In Section 4, we consider the continuous-time forward and backward process. Based on that, we were motivated to analyze the infinite limit of the average loss,

$$\lim_{t\to\infty} \frac{1}{T}\sum_{t=1}^{T} L_t\big(f, q(\mathbf{x}_t, \mathbf{x}_0)\big).$$

We find that the new loss can provide a better checkpoint than the loss averaged on the finite step on some tasks.

### B.3 CALCULATION OF THE EVIDENCE LOWER BOUND

### B.3.1 FINITE TIME DNDM

In this section, we derive the evidence lower bound (ELBO) for our model. The derivatives are inspired by the reasoning in DDIM (Song et al., 2020a). Specifically, We denote the generative process as $p_\theta(\mathbf{x}_{0:T}|\tau) = p_\theta^{(T)}(\mathbf{x}_T|\tau) \prod_{t=1}^T p_\theta^{(t)}(\mathbf{x}_{t-1}|\mathbf{x}_t, \tau)$. Here, $p_\theta^{(T)}$ is the pure noise and $p_\theta^{(t)}(\mathbf{x}_{t-1}|\mathbf{x}_t, \tau) = q(\mathbf{x}_{t-1}|\mathbf{x}_t, \widehat{\mathbf{x}}_0, \tau)$, where $\widehat{\mathbf{x}}_0$ is given by a neural network $f_\theta$, i.e., $\widehat{\mathbf{x}}_0 = f_\theta(\mathbf{x}_t, t)$. Notice that by Jensen's inequality,

$$\log p_\theta(\mathbf{x}_0) = \log \mathbb{E}_{\tau \sim \mathcal{D}_\tau}[p_\theta(\mathbf{x}_0|\tau)] \geq \mathbb{E}_{\tau \sim \mathcal{D}_\tau}[\log p_\theta(\mathbf{x}_0|\tau)]. \tag{13}$$

The evidence lower bound inequality gives

$$\log p_\theta(\mathbf{x}_0|\tau) \geq \mathbb{E}_{\mathbf{x}_{1:T} \sim q(\mathbf{x}_{1:T}|\mathbf{x}_0, \tau)} \log \frac{p_\theta(\mathbf{x}_{0:T}|\tau)}{q(\mathbf{x}_{1:T}|\mathbf{x}_0, \tau)}. \tag{14}$$

Plugging (14) into (13) gives the following ELBO,

$$\log p_\theta(\mathbf{x}_0) \geq \mathbb{E}_{\tau \sim \mathcal{D}_\tau} \mathbb{E}_{\mathbf{x}_{1:T} \sim q(\mathbf{x}_{1:T}|\mathbf{x}_0, \tau)} \log \frac{p_\theta(\mathbf{x}_{0:T}|\tau)}{q(\mathbf{x}_{1:T}|\mathbf{x}_0, \tau)} := \text{ELBO}.$$

We factorize the $p_\theta$ and $q$ by

$$p_\theta(\mathbf{x}_{0:T}|\tau) = p_\theta^{(T)}(\mathbf{x}_T|\tau) \prod_{t=1}^T p_\theta^{(t)}(\mathbf{x}_{t-1}|\mathbf{x}_t, \tau),$$

$$q(\mathbf{x}_{1:T}|\mathbf{x}_0, \tau) = q(\mathbf{x}_T|\mathbf{x}_0, \tau) \prod_{t=2}^T q(\mathbf{x}_{t-1}|\mathbf{x}_t, \mathbf{x}_0, \tau).$$

Here $q$ admits such a decomposition due to our definition of the diffusion process in (4), which introduce the following reverse process:

$$\mathbf{x}_{t-1} = \mathbb{1}(\tau = t)\mathbf{x}_0 + \mathbb{1}(\tau \neq t)\mathbf{x}_t.$$

Therefore, $\mathbf{x}_{1:T}$ is Markovian when conditioned on $\mathbf{x}_0$ and $\tau$. Based on the factorization, we have

$$\text{ELBO} = \mathbb{E}_{\tau \sim \mathcal{D}_\tau} \mathbb{E}_{\mathbf{x}_{1:T} \sim q(\mathbf{x}_{1:T}|\mathbf{x}_0, \tau)} \Big[ \log p_\theta^{(T)}(\mathbf{x}_T|\tau) + \sum_{t=1}^T \log p_\theta^{(t)}(\mathbf{x}_{t-1}|\mathbf{x}_t, \tau)$$

$$- \log q(\mathbf{x}_T|\mathbf{x}_0, \tau) - \sum_{t=2}^T \log q(\mathbf{x}_{t-1}|\mathbf{x}_t, \mathbf{x}_0, \tau) \Big]$$

$$= \mathbb{E}_{\tau \sim \mathcal{D}_\tau} \mathbb{E}_{\mathbf{x}_{1:T} \sim q(\mathbf{x}_{1:T}|\mathbf{x}_0, \tau)} \Big[ \log p_\theta^{(1)}(\mathbf{x}_0|\mathbf{x}_1, \tau) + \sum_{t=2}^T \log \frac{p_\theta^{(t)}(\mathbf{x}_{t-1}|\mathbf{x}_t, \tau)}{q(\mathbf{x}_{t-1}|\mathbf{x}_t, \mathbf{x}_0, \tau)}$$

$$+ \log \frac{p_\theta^{(T)}(\mathbf{x}_T|\tau)}{q(\mathbf{x}_T|\mathbf{x}_0, \tau)} \Big]$$

$$= \mathbb{E}_{\tau \sim \mathcal{D}_\tau} \mathbb{E}_{\mathbf{x}_1 \sim q(\cdot|\mathbf{x}_0, \tau)} \log p_\theta^{(1)}(\mathbf{x}_0|\mathbf{x}_1, \tau)$$

$$+ \sum_{t=2}^T \mathbb{E}_{\mathbf{x}_{t-1}, \mathbf{x}_t \sim q(\cdot|\mathbf{x}_0, \tau)} \log \frac{p_\theta^{(t)}(\mathbf{x}_{t-1}|\mathbf{x}_t, \tau)}{q(\mathbf{x}_{t-1}|\mathbf{x}_t, \mathbf{x}_0, \tau)} + \text{const}$$

$$= \mathbb{E}_{\tau \sim \mathcal{D}_\tau} \underbrace{\mathbb{E}_{\mathbf{x}_1 \sim q(\cdot|\mathbf{x}_0, \tau)} \log p_\theta^{(1)}(\mathbf{x}_0|\mathbf{x}_1, \tau)}_{\overline{\mathcal{L}}_1}$$

$$- \sum_{t=2}^T \mathbb{E}_{\tau \sim \mathcal{D}_\tau} \underbrace{\mathbb{E}_{\mathbf{x}_{t-1}, \mathbf{x}_t \sim q(\cdot|\mathbf{x}_0, \tau)} \text{KL}(q(\mathbf{x}_{t-1}|\mathbf{x}_t, \mathbf{x}_0, \tau)|p_\theta^{(t)}(\mathbf{x}_{t-1}|\mathbf{x}_t, \tau))}_{\overline{\mathcal{L}}_t} + \text{const}.$$

By a slight abuse of notations we use $q(\mathbf{x}_{t-1}|\mathbf{x}_t,\mathbf{x}_0), p_\theta^{(t)}(\mathbf{x}_0|\mathbf{x}_1)$ to indicate the distribution of the diffusion process defined in Zheng et al. (2023), that is, the standard Markov discrete diffusion process. In particular, we have

$$\overline{\mathcal{L}}_1 = \begin{cases} \mathbb{E}_{\mathbf{x}_1 \sim q(\cdot|\mathbf{x}_0)} \log p_\theta^{(1)}(\mathbf{x}_0|\mathbf{x}_1), & \tau = 1, \\ \text{const,} & \tau \neq 1. \end{cases}$$

$$\overline{\mathcal{L}}_t = \begin{cases} \mathbb{E}_{\mathbf{x}_{t-1},\mathbf{x}_t \sim q(\cdot|\mathbf{x}_0)} \text{KL}(q(\mathbf{x}_{t-1}|\mathbf{x}_t,\mathbf{x}_0)|p_\theta^{(t)}(\mathbf{x}_{t-1}|\mathbf{x}_t)), & \tau = t, \\ 0, & \tau \neq t. \end{cases}$$

Thus, we can obtain that

$$\text{ELBO} = \mathbb{P}(\tau = 1) \cdot \underbrace{\mathbb{E}_{\mathbf{x}_1 \sim q(\cdot|\mathbf{x}_0)} \log p_\theta^{(1)}(\mathbf{x}_0|\mathbf{x}_1)}_{\mathcal{L}_1}$$

$$- \sum_{t=2}^{T} \mathbb{P}(\tau = t) \cdot \underbrace{\mathbb{E}_{\mathbf{x}_{t-1},\mathbf{x}_t \sim q(\cdot|\mathbf{x}_0)} \text{KL}(q(\mathbf{x}_{t-1}|\mathbf{x}_t,\mathbf{x}_0)|p_\theta^{(t)}(\mathbf{x}_{t-1}|\mathbf{x}_t))}_{\mathcal{L}_t} + \text{const.}$$

Here $\mathcal{L}_t$ matches the loss terms in Zheng et al. (2023). In the practical training process, Zheng et al. (2023) samples $t$ from $\text{Unif}\{1, \cdots, T\}$ in each iteration and optimizes $\lambda_t \cdot \mathcal{L}_t$, where $\lambda_t$'s are weights. Thus, when we sample $\tau$ and optimize $\mathcal{L}_\tau$, our ELBO indeed leads to the same training objective as Zheng et al. (2023) up to reweighting. Since Zheng et al. (2023) is a parametrization of existing works (Austin et al., 2021; Hoogeboom et al., 2021b), our training objective indeed aligns with previous discrete diffusion models.

### B.3.2 CONTINUOUS TIME DNDM

In Section B.3, we derived an ELBO for DNDM and its accelerated algorithm defined in Section 2 and 3. While for finite sampling steps, we can decompose the diffusion process via the sampling steps $1, \ldots, T$ in (14), it becomes intractable for continuous Time DNDM (Infinite steps $T \to \infty$). Therefore, we will formulate the ELBO of continuous time DNDM by decomposing the transition times. The idea of decomposition of transition times follows Hoogeboom et al. (2021a), but their proof is only applicable to absorbing discrete diffusion, while ours can deal with discrete diffusion with various noise $q_{\text{noise}}$ including multinomial diffusion.

In Section B.3, we only consider the case of a single token $\mathbf{x} \in \mathbb{R}^K$ for simplicity as we decompose with the sampling steps $T$. In this Subsubsection, we will decompose over the transition time $\tau$. Therefore, we need to consider a sentence with multiple tokens $\mathbf{x}_{t,1:N} = [\mathbf{x}_{t,1}, \ldots, \mathbf{x}_{t,N}]$ where $\mathbf{x}_{t,n}$ is the $n$-th token and $N$ is the sequence length. Recall that we defined the transition time set $\mathcal{T} = \{\tau_n\}_{n=1}^{N}$ in Section 3. We arrange $\tau_n$ to obtain an ordered sequence $\tau_{n_k}$, where $0 = \tau_{n_0} < \tau_{n_1} < \tau_{n_2} < \ldots < \tau_{n_N} = T$. Then conditioning on the transition time set $\mathcal{T} = \{\tau_1, \ldots, \tau_N\}$, we have that

$$p_\theta(\mathbf{x}_{0:T,1:N}|\mathcal{T}) = p_\theta(\mathbf{x}_{\tau_{n_N},1:N}|\mathcal{T}) \prod_{s=N,\ldots,1} p_\theta(\mathbf{x}_{\tau_{n_{s-1}},1:N}|\mathbf{x}_{\tau_{n_s},1:N},\mathcal{T}),$$

where we omit the time superscript of $p$ for simplicity. Then the evidence lower bound inequality gives

$$\log p_\theta(\mathbf{x}_{0,1:N}|\mathcal{T}) \geq \mathbb{E}_{\mathbf{x}_{\tau_{n_1}:T,1:N} \sim q(\mathbf{x}_{\tau_{n_1}:T,1:N}|\mathbf{x}_{0,1:N},\mathcal{T})} \log \frac{p_\theta(\mathbf{x}_{0:T,1:N}|\mathcal{T})}{q(\mathbf{x}_{\tau_{n_1}:T,1:N}|\mathbf{x}_{0,1:N},\mathcal{T})}. \quad (15)$$

By Jensen's inequality, we have

$$\log p_\theta(\mathbf{x}_{0,1:N}) = \log \mathbb{E}_{\tau_1,\ldots,\tau_n \sim \mathcal{D}_\tau}[p_\theta(\mathbf{x}_{0,1:N}|\mathcal{T})] \geq \mathbb{E}_{\tau_1,\ldots,\tau_n \sim \mathcal{D}_\tau}[\log p_\theta(\mathbf{x}_0|\mathcal{T})]. \quad (16)$$

Plugging (15) into (16) gives the following ELBO,

$$\log p_\theta(\mathbf{x}_{0,1:N}) \geq \mathbb{E}_{\tau_1,\ldots,\tau_n \sim \mathcal{D}_\tau} \mathbb{E}_{\mathbf{x}_{\tau_{n_1}:T} \sim q(\mathbf{x}_{\tau_{n_1}:T}|\mathbf{x}_0,\mathcal{T})} \log \frac{p_\theta(\mathbf{x}_{0:T}|\mathcal{T})}{q(\mathbf{x}_{\tau_{n_1}:T}|\mathbf{x}_0,\mathcal{T})} := \text{ELBO}.$$

We factorize the $p_\theta$ and $q$ by

$$p_\theta(\mathbf{x}_{0:T,1:N}|\mathcal{T}) = p_\theta(\mathbf{x}_{T,1:N}|\mathcal{T}) \prod_{s=N,\ldots,1} p_\theta(\mathbf{x}_{\tau_{n_{s-1}},1:N}|\mathbf{x}_{\tau_{n_s},1:N},\mathcal{T}),$$

$$q(\mathbf{x}_{\tau_{n_1}:T,1:N}|\mathbf{x}_{0,1:N},\mathcal{T}) = q(\mathbf{x}_{T,1:N}|\mathbf{x}_0,\mathcal{T}) \prod_{s=N,\ldots,2} q(\mathbf{x}_{\tau_{n_{s-1}},1:N}|\mathbf{x}_{\tau_{n_s},1:N},\mathbf{x}_{0,1:N},\mathcal{T}).$$

Therefore, we have

$$
\begin{aligned}
\text{ELBO} = {}& \mathbb{E}_{\tau_1,\ldots,\tau_n \sim \mathcal{D}_\tau} \mathbb{E}_{\mathbf{x}_{\tau_{n_1}:T,1:N} \sim q(\mathbf{x}_{\tau_{n_1}:T,1:N}|\mathbf{x}_{0,1:N},\mathcal{T})} \Big[ \log p_\theta(\mathbf{x}_{T,1:N}|\mathcal{T}) \\
&+ \sum_{s=1}^{N} \log p_\theta(\mathbf{x}_{\tau_{n_{s-1}},1:N}|\mathbf{x}_{\tau_{n_s},1:N},\mathcal{T}) - \log q(\mathbf{x}_{T,1:N}|\mathbf{x}_{0,1:N},\mathcal{T}) \\
&- \sum_{s=2}^{N} \log q(\mathbf{x}_{\tau_{n_{s-1}},1:N}|\mathbf{x}_{\tau_{n_s},1:N},\mathbf{x}_{0,1:N},\mathcal{T}) \Big] \\
= {}& \mathbb{E}_{\tau_1,\ldots,\tau_n \sim \mathcal{D}_\tau} \mathbb{E}_{\mathbf{x}_{\tau_{n_1}:T,1:N} \sim q(\mathbf{x}_{\tau_{n_1}:T,1:N}|\mathbf{x}_{0,1:N},\mathcal{T})} \Big[ \log p_\theta(\mathbf{x}_{0,1:N}|\mathbf{x}_{1,1:N},\mathcal{T}) \\
&+ \sum_{s=2}^{N} \log \frac{p_\theta(\mathbf{x}_{\tau_{n_{s-1}},1:N}|\mathbf{x}_{\tau_{n_s},1:N},\mathcal{T})}{q(\mathbf{x}_{\tau_{n_{s-1}},1:N}|\mathbf{x}_{\tau_{n_s},1:N},\mathbf{x}_{0,1:N},\mathcal{T})} + \log \frac{p_\theta(\mathbf{x}_{T,1:N}|\mathcal{T})}{q(\mathbf{x}_{T,1:N}|\mathbf{x}_{0,1:N},\mathcal{T})} \Big] \\
= {}& \mathbb{E}_{\tau_1,\ldots,\tau_n \sim \mathcal{D}_\tau} \mathbb{E}_{\mathbf{x}_{1,1:N} \sim q(\cdot|\mathbf{x}_{0,1:N},\mathcal{T})} \log p_\theta(\mathbf{x}_{0,1:N}|\mathbf{x}_{1,1:N},\mathcal{T}) \\
&+ \sum_{s=2}^{N} \mathbb{E}_{\mathbf{x}_{\tau_{n_{s-1}},1:N},\mathbf{x}_{\tau_{n_s},1:N} \sim q(\cdot|\mathbf{x}_{0,1:N},\mathcal{T})} \log \frac{p_\theta(\mathbf{x}_{\tau_{n_{s-1}},1:N}|\mathbf{x}_{\tau_{n_s},1:N},\mathcal{T})}{q(\mathbf{x}_{\tau_{n_{s-1}},1:N}|\mathbf{x}_{\tau_{n_s},1:N},\mathbf{x}_{0,1:N},\mathcal{T})} + \text{const} \\
= {}& \mathbb{E}_{\tau_1,\ldots,\tau_n \sim \mathcal{D}_\tau} \mathbb{E}_{\mathbf{x}_{1,1:N} \sim q(\cdot|\mathbf{x}_{0,1:N},\mathcal{T})} \log p_\theta(\mathbf{x}_{0,1:N}|\mathbf{x}_{1,1:N},\mathcal{T}) \\
&- \sum_{s=2}^{N} \mathbb{E}_{\tau_1,\ldots,\tau_n \sim \mathcal{D}_\tau} \mathbb{E}_{\mathbf{x}_{\tau_{n_{s-1}},1:N},\mathbf{x}_{\tau_{n_s},1:N} \sim q(\cdot|\mathbf{x}_{0,1:N},\mathcal{T})} \\
&\quad \text{KL}(q(\mathbf{x}_{\tau_{n_{s-1}},1:N}|\mathbf{x}_{\tau_{n_s},1:N},\mathbf{x}_{0,1:N},\mathcal{T})|p_\theta(\mathbf{x}_{\tau_{n_{s-1}},1:N}|\mathbf{x}_{\tau_{n_s},1:N},\mathcal{T})) + \text{const}. \qquad (17)
\end{aligned}
$$

*Remark* B.1. (17) represents the ELBO utilized by the DNDM-C architecture. As our transition times $\tau_n$ are independently and identically drawn from the distribution $\mathcal{D}_\tau$, we are unable to further decompose (17) into a loss function related to the position information $1 : N$, as was accomplished by Hoogeboom et al. (2021a).

## C CHOICE OF THE TRANSITION TIME

Transition time $\tau$ in Definition 2.2 plays an important role in DNDM. In this section, we provide a deeper discussion of the transition time. We first give a proof of the Theorem 3.1.

*Proof of Theorem 3.1.* By the definition of $\tau$, we know that $\tau_n = t$ is equivalent to $b_{0,n} = 1, \ldots, b_{t-1,n} = 1$ and $b_{t,n} = 0$. Since $\{b_{t,n}\}_{t=0}^{T}$ is independent for different $n$ by definition, each $\tau_n$ is also independent. Therefore, we drop the subscript $n$ for simplicity. On the other hand if $b_0 = 1, \ldots, b_{t-1} = 1$ and $b_t = 0$ we can also conclude that $\tau = t$. Therefore, we have that

$$
\begin{aligned}
\mathbb{P}(\tau = t) &= \mathbb{P}(b_0 = 1, \ldots, b_{t-1} = 1, b_t = 0) \\
&= \left[ \Pi_{s=1}^{t-1} \beta_s \right] \cdot (1 - \beta_t) \\
&= \Pi_{s=1}^{t-1} \beta_s - \Pi_{s=1}^{t} \beta_s \\
&= \alpha_{t-1} - \alpha_t,
\end{aligned}
$$

where the second equality is due to $b_s, s = 1, 2, \ldots, t$ are independent random variable following Bernoulli($\beta_s$) distribution and the last equality is by the definition of $\alpha_t = \Pi_{s=1}^{t} \beta_s$. $\square$

Notice that $\alpha_t$ is a decreasing sequence in the 0 to 1 range. Therefore, $\mathbb{P}(\tau = t) \in [0, 1]$ for any $t \in \{1, \ldots, T\}$. Besides $\sum \mathbb{P}(\tau = t) = \sum_{t=1}^{T} (\alpha_{t-1} - \alpha_t) = \alpha_0 - \alpha_T = 1$. Therefore, the derived distribution is valid as long as the $\alpha_t$ is decreasing from 1 to 0.

From Theorem 3.1, we discern that the nature of the diffusion model scheduler, $\alpha_t$, clarifies the distribution of $\tau$.

**Linear $\alpha$ schedule.** This is a schedule studied in (Austin et al., 2021), where $\alpha_t = 1 - t/T$. This will result in $\mathbb{P}(\tau_n = t) = 1/T$ for every $t$ in the range 1 to $T$. As a result, transition time distributes uniformly across each moment in the set $\{1, \ldots, T\}$. This can be verified in a) of Figure 3.

**Cosine $\alpha$ schedule.** This is a schedule studied in (Hoogeboom et al., 2021b), where $\alpha_t = \cos(\pi * t/2T)$. For numerical consideration of the noise, a small offset $s$ is added, i.e., $\alpha_t = f(t)/f(0)$

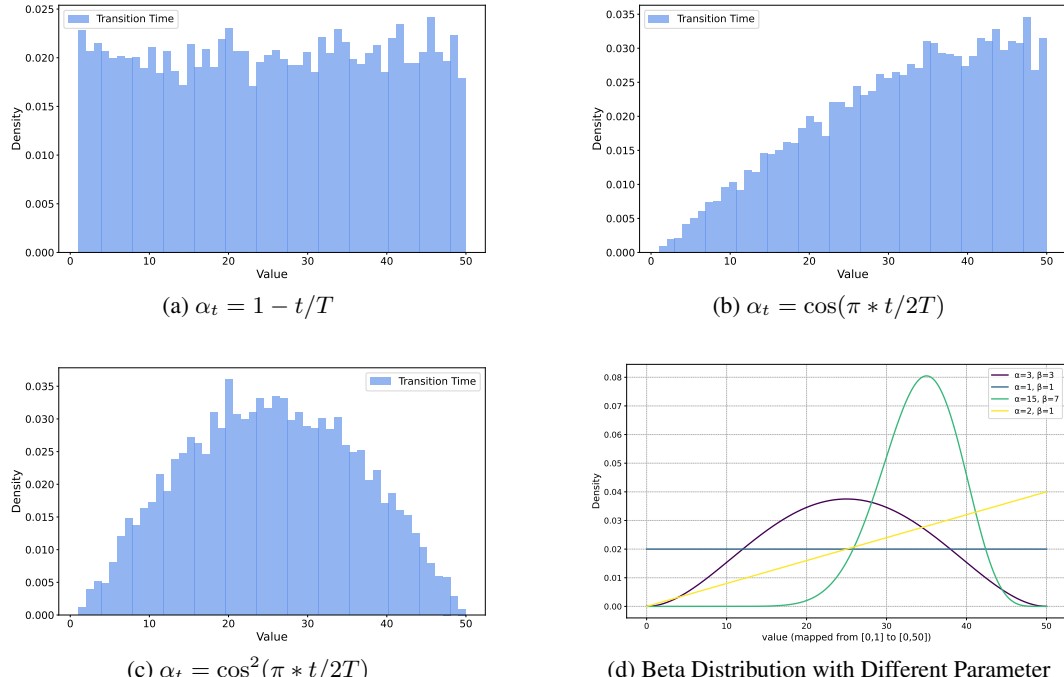

Figure 3: Different distribution of transition time for $T = 50$. $a), b), c)$ The transition time sampled 1K times under the different $\alpha_t$ schedule. d) The approximated transition time for $t = 1, \ldots, T$ using different hypter-parameters.

where $f(t) = \cos((s + t/T)/(1 + s) * \pi/2)$. As shown in b) of Figure 3, the transition time will concentrate more on the large $T$.

**Cosine square $\alpha$ schedule.** This is a schedule studied in (Zheng et al., 2023), where $\alpha_t = \cos^2(\pi * t/2T)$, which motivated by Nichol & Dhariwal (2021). Again, for numerical consideration of the noise, a small offset $s$ is added, i.e., $\alpha_t = f(t)/f(0)$ where $f(t) = \cos^($ $(s + t/T)/(1 + s) * \pi/2)$. As shown in c) of Figure 3, the transition time will concentrate more on the middle of the range.

Generally, if we express $\alpha_t$ as $g(t/T)$, then we can simplify to $\mathbb{P}(\tau = t) = g((t - 1)/T) - g(t/T)$, which further refines to $(1/T)|g'(t/T)| + o(1/T)$. This indicates that transitions are more likely where $|g'|$ is large. Such a mathematical finding can match our observation in Figure 3.

In practice, we find that the shape of the transition time doesn't need to match the theoretical prediction schedule exactly. As we can see from d) in Figure 3. A reshaped Beta distribution can approximate all the transition time distributions in a fixed range. We first extract a time $t \in [0, 1]$ from a Beta distribution, then adjust these samples to fit by multiplying $T$ and round them to acquire the integer. Our experiment finds that a properly chosen Beta distribution (tuned on the validation set) makes DNDM perform better on the translation tasks. Specifically, the chosen Beta distributions and the searching method are reported in Appendix F. The performance of the four transition time schedules mentioned above, including the reported Beta distributions for comparison, are listed in Table 4, where we find the other three schedules affect the performance and most of their scores are lower than the scores of Beta distribution, but their scores are at least still close to the reported Beta distributions, especially for DNDM-k-absorb and DNDM-absorb. The efficiencies (measured by NFE) are also similar to one another.

Additionally, the ablation study on a reasonable range of different Beta distributions with 50 and 1000 sampling steps is shown in Tables 8 and 7, where the BLEU scores and NFE values on the test set of one of the three machine translation datasets, WMT16, are shown for demonstration. The range of Beta distributions covers our chosen Beta schedules based on validation sets and a variety of basic Beta distribution shapes. These results show that the different Beta distributions influence the performance, but most of these choices of parameters still achieve results close to the optimal. Since the Beta distributions of the reported results in Tables 1 and 2 are selected using the validation set, they do not always have the highest scores on the test set, but their scores still at least belong to the top tiers according to these tables.

**Another view of the transition time.** In Algorithm 1, we only need to call the neural network when $t \in \mathcal{T}$, which can significantly speed up the sampling since we reduce the function call. Notice that after we get the $\mathbf{x}_0$ prediction, we only update the $\mathbf{x}_t$ for those tokens at the transition time. However, (5) implies that $\mathbf{x}_t = \mathbf{x}_0$ as long as $\tau > t$. Therefore, instead of only updating the $\mathbf{x}_t$ for those tokens at the transition time, i.e., $\tau = t$, we can also update those tokens with transition time $\tau >= t$. This motivates us to consider a variation presented as Algorithm 3, which keeps almost the same sampling time but will update the tokens several times rather than just once. Since the tokens now get the chance to be corrected over time. The new Algorithm 3 will be more robust than Algorithm 1.

| Datasets | Schedules | DNDM-multi | | DNDM-absorb | | DNDM-k-multi | | DNDM-k-absorb | |
|---|---|---|---|---|---|---|---|---|---|
| | | BLEU | Avg NFE | BLEU | Avg NFE | BLEU | Avg NFE | BLEU | Avg NFE |
| IWSLT14 | Cosine | 31.72 | 31.71 | 32.71 | 31.21 | 32.91 | 31.71 | 34.50 | 31.21 |
| | Cosine$^2$ | 31.78 | 31.74 | **32.93** | 31.21 | 32.78 | 31.74 | 34.53 | 31.21 |
| | Linear $\alpha$ | 31.77 | 31.82 | 32.65 | 31.33 | 32.83 | 31.82 | 34.53 | 31.33 |
| | Beta (reported) | **31.82** | **30.33** | **32.93** | **31.08** | **33.15** | **30.33** | **34.56** | **31.08** |
| WMT14 | Cosine | **25.80** | 39.61 | **26.54** | 39.18 | 26.63 | 39.61 | 27.81 | 39.18 |
| | Cosine$^2$ | 25.52 | 39.48 | 26.53 | 39.18 | 25.01 | 39.48 | **27.95** | 39.18 |
| | Linear $\alpha$ | 25.58 | 39.97 | 26.33 | 39.82 | 25.47 | 39.97 | 27.63 | 39.82 |
| | Beta (reported) | 25.71 | **38.94** | 26.43 | **38.76** | **26.88** | **38.94** | 27.82 | **38.76** |
| WMT16 | Cosine | 32.71 | 40.50 | 33.56 | 40.45 | 33.46 | 40.50 | 34.37 | 40.45 |
| | Cosine$^2$ | 32.73 | 40.50 | 33.51 | 40.45 | 33.44 | 40.50 | 34.24 | 40.45 |
| | Linear $\alpha$ | 32.85 | 40.36 | 33.46 | 40.36 | 33.47 | 40.36 | 33.88 | 40.36 |
| | Beta (reported) | **32.86** | **38.46** | **33.60** | **38.27** | **33.79** | **38.45** | **34.38** | **38.27** |

Table 4: The BLEU scores and average number of function evaluations (NFE) values of different distributions of transition time for 1000 sampling steps with batch size 100. The parameters of the Beta distributions in this table are the same as in Tables 1 and 2 and are reported in Appendix F.

## D    DISCUSSION OF THE NUMBER OF FUNCTION EVALUATIONS (NFE).

In this section, we discuss the number of function evaluations (NFE) in DNDM. According to (7), the update of a token $\mathbf{x}_{t-1,n}$ occurs solely at its designated transition time. Meanwhile, if step $t$ does not coincide with a transition time for any token, we maintain the sentence from the preceding step unchanged: $\mathbf{x}_{t,1:N} = \mathbf{x}_{t-1,1:N}$. Therefore, our algorithm removes the need of evaluating $f_n(\mathbf{x}_{t,1:N}, t)$ for steps outside the set of transition times. Given this structure, our analytical emphasis is on the transition set $\mathcal{T}$ since function evaluations are required only at times $t$ that are members of $\mathcal{T}$. Consequently, the NFE is precisely the cardinality of the transition set, denoted as $|\mathcal{T}|$. In our main paper, we propose a naive upper bound for $|\mathcal{T}|$ as $\min\{N, T\}$, which effectively demonstrates the efficiency of our method when $T > N$. Next, we demonstrate that DNDM also reduces the NFE when $T < N$, by providing a precise estimation of $|\mathcal{T}|$.

**Theorem D.1.** *Suppose transition time follows distribution $\mathcal{D}_\tau$, and consider a sequence of length $N$. Then, the cardinality of the transition set $\mathcal{T} := \{\tau_1, \ldots, \tau_N\}$ satisfies:*
- $1 \leq |\mathcal{T}| \leq \min\{N, T\}$,
- $\mathbb{E}[|\mathcal{T}|] = [1 - C_{T,N,\mathcal{D}_\tau}] \cdot T$, *where $C_{T,N,\mathcal{D}_\tau}$ is a constant in the range $(0, 1)$. Furthermore,*

$$C_{T,N,\mathcal{D}_\tau} = \Big( \sum_{i=1}^{T} (1 - p_i)^N \Big)/T \geq (1 - 1/T)^N,$$

*where $p_i = \mathbb{P}(\tau = i)$ for $\tau \sim \mathcal{D}_\tau$, and the equality holds if and only if $\mathcal{D}_\tau$ is a uniform distribution.*

*Proof.* The first statement is straightforward. For completeness, the proof is provided. Since there are only $N$ transition times (possibly repeated): $\tau_1, \ldots, \tau_N$, the distinct transition times must satisfy $|\mathcal{T}| \leq N$. Additionally, since $\mathcal{T} \subseteq \{1, \ldots, T\}$, we also have $|\mathcal{T}| \leq T$.

To prove the second statement, we decompose $\mathcal{T}$ and use the property of expectation. Note that $|\mathcal{T}| = \sum_{i=1}^{T} \mathbb{1}\{i \in \mathcal{T}\}$. Thus,

$$\mathbb{E}[|\mathcal{T}|] = \mathbb{E}\Big[ \sum_{i=1}^{T} \mathbb{1}\{i \in \mathcal{T}\} \Big] = \sum_{i=1}^{T} \mathbb{P}(i \in \mathcal{T}). \tag{18}$$

Assuming $\mathbb{P}_{\mathcal{D}_\tau}(\tau = i) = p_i$, and that $\tau_n$ are i.i.d. draws from $\mathcal{D}_\tau$, we have

$$\mathbb{P}(i \in \mathcal{T}) = 1 - \mathbb{P}(i \notin \mathcal{T}) = 1 - (1 - p_i)^N. \tag{19}$$

Substituting (19) into (18) yields

$$\mathbb{E}[|\mathcal{T}|] = \sum_{i=1}^{T} \left[ 1 - (1 - p_i)^N \right] = \left[ 1 - \frac{\sum_{i=1}^{T}(1 - p_i)^N}{T} \right] \cdot T = [1 - C_{T,N,\mathcal{D}_\tau}] \cdot T,$$

where $C_{T,N,\mathcal{D}_\tau} = \left( \sum_{i=1}^{T}(1 - p_i)^N \right)/T$. An upper bound for $C_{T,N,\mathcal{D}_\tau}$ is given as

$$C_{T,N,\mathcal{D}_\tau} = \left[ 1 - \frac{\sum_{i=1}^{T}(1 - p_i)^N}{T} \right] \cdot T$$
$$\leq \left[ 1 - \left( 1 - \frac{1}{T} \right)^N \right] \cdot T,$$

where the inequality holds if and only if $p_i = 1/T$ for all $i \in [T]$, i.e., $\mathcal{D}_\tau$ is a uniform distribution. $\square$

*Remark* D.2. Theorem D.1 suggests that even when $T \leq N$, our method still provides a significant improvement. Specifically, for $T = N \geq 4$, we have $C_{T,N,\mathcal{D}_\tau} = (1 - 1/N)^N \geq 0.3$. This implies that our model requires at most $0.7T$ even in the worst case. Moreover, if we consider a special scenario where the number of $p_i$ satisfying $p_i < \epsilon$ is more than $M$, then we have $C_{T,N,\mathcal{D}_\tau} > M(1 - \epsilon)^N/T$, indicating that with $M$ sufficiently large and $\epsilon$ sufficiently small, $C_{T,N,\mathcal{D}_\tau}$ can be pretty close to 1.

*Remark* D.3. In practical applications of our model, we employ a beta distribution for $\mathcal{D}_\tau$, which typically exhibits a right-heavy tail. Therefore $C_{T,N,\mathcal{D}_\tau}$ tends to be larger than that in the worst-case scenario. In Tables 5 and 6, we list the average NFE for each experiment we run in §5. These results demonstrate a significant reduction in NFE compared to the original counts: for $T = 25$, the NFE is only about half of the original count; for $T = 50$, it is approximately one-third; and for $T = 1000$, it reduces to less than one-twentieth of the original count.

*Remark* D.4. By Bernoulli's inequality, $(1 - p)^N > 1 - N \cdot p$ for $1 > p > 0$. Therefore, $C_{T,N,\mathcal{D}_\tau} > 1 - N/T$, implying that $\mathbb{E}[|\mathcal{T}|] < N$. As $T \to \infty$, assuming the transition time does not concentrate at a single point, the probability that two transitions occur simultaneously is zero. Consequently, the generation process will sequentially go through each token. Thus, the expected number of function evaluations (NFE), $\mathbb{E}[|\mathcal{T}|]$, will be $N$. In contrast, when $T$ is finite, there is a non-zero probability that multiple transitions happen at the same time. Hence, in this case, the NFE, $|\mathcal{T}|$, is strictly less than $N$

# E   DISCRETE NON-MARKOV DIFFUSION MODEL WITH TOP-K TRANSITION TIME (DNDM-K).

Recent works have demonstrated that the quality of samples can be enhanced by utilizing supplementary information derived from the neural network, $f_\theta(\cdot, t)$ (Ghazvininejad et al., 2019; Savinov et al., 2021; Chang et al., 2022; He et al., 2022). Very recently, Zheng et al. (2023) applied this idea in their RDM framework and can achieve significant performance improvement. Specifically, after decoding $\widehat{\mathbf{x}}_{0,1:N}$ from transformer $f_\theta(\mathbf{x}_{t,1:N}, t)$, the score corresponding to this decoded token from the transformer's last layer, is also recorded and denote as $s_{t,n}$. Tokens with high scores are more likely to be selected for updates.

Inspired by Zheng et al. (2023), we introduce the discrete non-Markov discrete diffusion Model with top-K transition time (DNDM-K). Instead of directly determining which token gets updated at step $t$ by first drawing transition time $\tau \sim \mathcal{D}_\tau$, we employ a two-step process.

1. We first compute $K_t = \sum_{n=1}^{N} \mathbb{1}(\tau_n \geq t)$. $k_t$ represents how many tokens should be decoded at the current step.
2. Compare $K_{t-1}$ and $K_t$, if $K_{t-1} = K_t$. There is no transition time at time $t$, we just update $\mathbf{x}_{t-1,1:N} = \mathbf{x}_{t,1:N}$. If $K_{t-1} > K_t$, Then there exist transition time at time $t$, we calculate and select the indexes with top-$K_{t-1}$ scores. Then we update those tokens if it hasn't been updated yet.

Subsequently, we will only update those tokens with the highest $K_t$ score that hasn't been changed yet. Since the function evaluation occurs only when $K_t$ changes, DNDM-K can give an accelerated sampling algorithm. The details are presented in Algorithm 4.

| Dataset | Steps | RDM-Multi | | DNDM-Multi | | RDM-$k$-Multi | | DNDM-$k$-Multi | |
|---|---|---|---|---|---|---|---|---|---|
| | | BLEU | Avg NFE | BLEU | Avg NFE | BLEU | Avg NFE | BLEU | Avg NFE |
| IWSLT14 | 25 | **31.26** | 25 | 30.95 | **9.03** | **32.82** | 25 | 32.30 | **9.03** |
| | 50 | **31.50** | 50 | 31.45 | **14.07** | **32.82** | 50 | 32.80 | **14.07** |
| | 1000 | 31.69 | 1000 | **31.82** | **30.33** | 32.64 | 1000 | **33.15** | **30.33** |
| | $\infty$ | - | - | **31.89** | **32.73** | - | - | **33.44** | **32.73** |
| WMT14 | 25 | **25.25** | 25 | 25.01 | **13.52** | 26.03 | 25 | 25.98 | **13.52** |
| | 50 | **25.75** | 50 | 25.33 | **20.58** | 26.14 | 50 | **26.37** | **20.58** |
| | 1000 | 25.66 | 1000 | **25.71** | **38.94** | 25.82 | 1000 | **26.88** | **38.94** |
| | $\infty$ | - | - | **24.79** | **40.67** | - | - | **26.39** | **40.67** |
| WMT16 | 25 | **32.29** | 25 | 31.97 | **8.5** | 33.12 | 25 | 32.94 | **8.5** |
| | 50 | **32.53** | 50 | 32.50 | **14.73** | 33.41 | 50 | 33.26 | **14.73** |
| | 1000 | 32.63 | 1000 | **32.86** | **38.45** | 33.67 | 1000 | **33.79** | **38.45** |
| | $\infty$ | - | - | **32.91** | **41.64** | - | - | **33.86** | **41.64** |

Table 5: BLEU score and the average number of function evaluations (NFE) comparison of multinomial diffusion on machine translation benchmarks IWSLT14 DE-EN, WMT14 EN-DE, and WMT16 EN-RO. The blue background highlights our algorithms. The average NFE values are calculated by dividing the number of times calling the denoising function (neural network) during generation by the number of batches, where the batch sizes of all experiments are 100.

| Dataset | Steps | RDM-Absorb | | DNDM-Absorb | | RDM-$k$-Absorb | | DNDM-$k$-Absorb | |
|---|---|---|---|---|---|---|---|---|---|
| | | BLEU | Avg NFE | BLEU | Avg NFE | BLEU | Avg NFE | BLEU | Avg NFE |
| IWSLT14 | 25 | 31.58 | 25 | **32.43** | **13.81** | **34.50** | 25 | 34.14 | **13.81** |
| | 50 | 31.80 | 50 | **32.63** | **19.24** | **34.58** | 50 | 34.34 | **19.24** |
| | 1000 | 31.91 | 1000 | **32.93** | **31.08** | **34.60** | 1000 | 34.56 | **31.08** |
| | $\infty$ | - | - | **33.03** | **32.07** | - | - | **34.65** | **32.07** |
| WMT14 | 25 | 24.97 | 25 | **25.79** | **15.09** | **27.50** | 25 | 27.18 | **15.09** |
| | 50 | 24.95 | 50 | **26.10** | **22.45** | **27.73** | 50 | 27.66 | **22.45** |
| | 1000 | 25.22 | 1000 | **26.43** | **38.76** | 27.75 | 1000 | **27.82** | **38.76** |
| | $\infty$ | - | - | **26.50** | **40.39** | - | - | **27.50** | **40.39** |
| WMT16 | 25 | 32.86 | 25 | **33.20** | **13.91** | 33.92 | 25 | **33.96** | **13.91** |
| | 50 | 32.93 | 50 | **33.30** | **20.95** | 34.10 | 50 | **34.20** | **20.95** |
| | 1000 | 33.25 | 1000 | **33.60** | **38.27** | **34.44** | 1000 | 34.38 | **38.27** |
| | $\infty$ | - | - | **33.42** | **41.59** | - | - | **34.41** | **41.59** |

Table 6: BLEU score and the average number of function evaluations (NFE) comparison of absorbing diffusion on machine translation benchmarks IWSLT14 DE-EN, WMT14 EN-DE, and WMT16 EN-RO. The blue background highlights our algorithms. The average NFE values are calculated by dividing the number of times calling the denoising function (neural network) during generation by the number of batches, where the batch sizes of all experiments are 100.

## F  EXPERIMENT DETAILS

### F.1  CONDITIONAL TEXT GENERATION

**Parameter choices.** In all experiments, the batch size is chosen to be 100. For RDM and RDM-$k$, our hyperparameter settings follow the original paper (Zheng et al., 2023) except for the batch size. Before the sampling, we used the saved checkpoint of trained models provided by the authors for discrete sampling experiments, and we trained the corresponding models for continuous sampling experiments.

For finite-step DNDM, the transition times are determined by the schedule, and we approximate the schedule with a Beta distribution Beta$(\alpha, \beta)$ (please refer to Section 3 for detailed explanation). The $\alpha$ and $\beta$ values are selected by applying grid search on the validation sets. Based on the BLEU scores on the validation sets, we have selected Beta$(15, 7)$ for Multinormal Diffusion on IWSLT14, Beta$(3, 3)$ for Absorbing Diffusion on both IWSLT14 and WMT14, Beta$(5, 3)$ for Multinormal Diffusion on WMT14 and Absorbing Diffusion on WMT16, and Beta$(20, 7)$ for Multinormal Diffusion on WMT16.

**Algorithm 3** Sampling From DNDM (Version 2)

**Require:** Trained prediction function $f_\theta(\cdot, t)$, $\mathbf{q}_{\text{noise}}$, $\mathcal{D}_\tau$
1: **for** $n = 1 \ldots N$ **do**
2:     Initiate each token $\mathbf{x}_{T,n} \sim \mathbf{q}_{\text{noise}}$
3:     Initiate the transition time $\tau_n \sim \mathcal{D}_\tau$
4: **end for**
5: Collect transition time set $\mathcal{T} = \{\tau_n\}_{n=1}^N$
6: **for** $t = T \ldots 1$ **do**
7:     **if** $t \in \mathcal{T}$ **then**
8:         Generate $\widetilde{\mathbf{x}}_{0,1:N}$ from $f_\theta(\mathbf{x}_{t,1:N}, t)$
9:         **for** $n = 1 \ldots N$ **do**
10:            Update $\mathbf{x}_{t-1,n}$ if $\tau_n \geq t$
11:         **end for**
12:     **else**
13:         Update $\mathbf{x}_{t-1,1:N} = \mathbf{x}_{t,1:N}$
14:     **end if**
15: **end for**
16: **return** $\mathbf{x}_{0,1:N}$

**Algorithm 4** Sampling From DNDM-K

**Input:** Trained prediction function $f_\theta(\cdot, t)$, $\mathbf{q}_{\text{noise}}$ and $\mathcal{D}_\tau$
**for** $n = 1 \ldots N$ **do**
    Initiate each token $\mathbf{x}_{T,n} \sim \mathbf{q}_{\text{noise}}$
    Initiate the top K number $\{K_t\}$
    Initiate an empty set $U = \{\}$, which includes the index of the tokens that have been updated.
**end for**
**for** $t = T \ldots 1$ **do**
    **if** $K_{t-1} > K_t$ **then**
        Calculate the $\mathcal{P} = \text{argtop}_{K_t}\{s_{t,n}\}_{n=1}^N$;
        Generate $\widetilde{\mathbf{x}}_{0,1:N}$ from $f_\theta(\mathbf{x}_{t,1:N}, t)$
        Update $\mathbf{x}_{t-1,n} = \widetilde{\mathbf{x}}_{0,n}$ for all $n$ in the set $\mathcal{P}$ but not in the set $U$ (top score but not updated yet)
        Update the set $U$ by appending the index of the updated tokens
    **else**
        Update $\mathbf{x}_{t-1,1:N} = \mathbf{x}_{t,1:N}$;
    **end if**
**end for**
**return** $\mathbf{x}_{0,1:N}$.

For infinite-steps (continuous-step) diffusion (DNDM-C), the transition timestamps are sampled from $\text{Beta}(\alpha, \beta)$, where the choice of $(\alpha, \beta)$ are chosen from $(100.0, 4.0)$ or $(17.0, 4.0)$, based on the performance comparison on the validation set. In the end we choose $\text{Beta}(17, 4)$ for IWSLT14 and $\text{Beta}(100, 4)$ for WMT14 and WMT16.

We conduct a performance comparison based on varying configurations of the Beta and Alpha distributions. The results of these comparisons are presented in Tables 8 and 7. Furthermore, to evaluate the efficacy of discrete versus continuous step schemes, we also conduct an ablation study under the same set of parameters $(100, 4)$ in Table 9.

| Model | Alpha | Beta | | | | | | | | | |
|---|---|---|---|---|---|---|---|---|---|---|---|
| | | 3 | 5 | 7 | 9 | 11 | 13 | 15 | 17 | 19 | 21 |
| DNDM-k-Multi | 3 | 33.47 | 33.67 | 33.62 | 33.77 | **33.87** | 33.64 | 33.73 | 33.60 | 33.68 | 33.56 |
| | 5 | 33.18 | 33.47 | 33.68 | 33.53 | 33.71 | 33.69 | 33.73 | 33.72 | 33.74 | 33.82 |
| | 7 | 32.99 | 33.20 | 33.49 | 33.56 | 33.58 | 33.61 | 33.67 | 33.72 | 33.78 | 33.83 |
| DNDM-Multi | 3 | 32.73 | 32.66 | 32.74 | 32.82 | 32.77 | **32.92** | 32.80 | 32.81 | 32.76 | 32.86 |
| | 5 | 32.32 | 32.62 | 32.70 | 32.80 | 32.83 | 32.83 | 32.90 | 32.95 | 32.91 | 32.87 |
| | 7 | 32.35 | 32.35 | 32.53 | 32.67 | 32.75 | 32.78 | 32.86 | 32.80 | 32.86 | 32.88 |
| DNDM-k-Absorb | 3 | 34.19 | 34.38 | 34.34 | 34.22 | 34.21 | 34.24 | 34.07 | 34.31 | **34.42** | 34.36 |
| | 5 | 32.15 | 33.99 | 34.29 | 34.30 | 34.29 | 34.40 | 34.40 | 34.24 | 34.30 | 34.22 |
| | 7 | 27.67 | 32.87 | 33.94 | 34.28 | 34.27 | 34.38 | 34.31 | 34.29 | 34.38 | 34.40 |
| DNDM-Absorb | 3 | 33.53 | 33.60 | 33.67 | 33.71 | 33.71 | 33.70 | 33.58 | 33.63 | 33.53 | 33.54 |
| | 5 | 32.70 | 33.33 | 33.52 | 33.60 | 33.66 | 33.73 | 33.70 | **33.74** | 33.72 | **33.74** |
| | 7 | 30.56 | 32.65 | 33.28 | 33.37 | 33.51 | 33.52 | 33.61 | 33.67 | 33.63 | 33.67 |

Table 7: BLEU scores on dataset WMT16 from the ablation study of other different $\text{Beta}(\alpha, \beta)$ distributions of the transition time with 1000 sampling steps.

**Continuous time vs discrete time diffusions.** To test our hypothesis that the continuous-time sampler will produce more accurate results in reverse sampling if our $\mathbf{x}_0$ estimator consistently approximates the true $\mathbf{x}_0$ over time, we conduct various sampling experiments using a shared pre-trained neural network. For discrete-time sampling, we consider three cases: $T = 25, 50, 1000$. In each case, we rescale the interval $[0, T]$ to $[0, 50]$ and divide it into $T$ fractions. In contrast, for continuous-time sampling, we directly sample from a continuous distribution over the interval $[0, 50]$ without any partitioning.

| Model | Alpha | Beta | | | | | | | | | |
|---|---|---|---|---|---|---|---|---|---|---|---|
| | | 3 | 5 | 7 | 9 | 11 | 13 | 15 | 17 | 19 | 21 |
| DNDM-k-Multi | 3 | 33.31 | 33.47 | 33.39 | 33.48 | 33.29 | 33.23 | 33.25 | 33.27 | 33.11 | 33.17 |
| | 5 | 32.93 | 33.28 | 33.29 | **33.58** | 33.45 | 33.21 | 33.40 | 33.49 | 33.16 | 33.19 |
| | 7 | 32.61 | 32.98 | 33.31 | 33.20 | 33.27 | 33.41 | 33.39 | 33.53 | 33.35 | 33.08 |
| DNDM-Multi | 3 | 32.63 | 32.46 | 32.44 | 32.56 | 32.59 | 32.55 | 32.37 | 32.33 | 32.22 | 32.23 |
| | 5 | 32.31 | 32.43 | 32.66 | 32.64 | **32.68** | 32.55 | 32.55 | 32.44 | 32.35 | 32.30 |
| | 7 | 31.95 | 32.11 | 32.22 | 32.26 | 32.54 | 32.52 | 32.50 | 32.58 | 32.48 | 32.41 |
| DNDM-k-Absorb | 3 | 34.05 | 34.2 | 34.31 | 34.37 | 34.15 | 34.05 | 34.06 | 33.77 | 33.81 | 33.84 |
| | 5 | 32.30 | 34.08 | 34.30 | **34.38** | 34.26 | 34.23 | 34.09 | 34.06 | 34.02 | 34.13 |
| | 7 | 27.39 | 32.64 | 33.71 | 34.18 | 34.02 | 34.33 | 34.31 | 34.17 | 34.12 | 34.19 |
| DNDM-Absorb | 3 | 33.26 | 33.30 | 33.29 | 33.24 | 33.23 | 32.97 | 33.06 | 32.85 | 32.89 | 32.63 |
| | 5 | 32.47 | 33.08 | 33.31 | 33.22 | **33.41** | 33.25 | 33.15 | 33.27 | 33.04 | 32.98 |
| | 7 | 30.34 | 32.27 | 33.27 | 33.03 | 33.16 | 33.14 | 33.27 | 33.11 | 33.11 | 33.07 |

Table 8: BLEU scores on dataset `WMT16` from the ablation study of other different Beta$(\alpha, \beta)$ distributions of the transition time with 50 sampling steps.

| Steps | DNDM-k-multi | DNDM-k-absorb | DNDM-multi | DNDM-absorb |
|---|---|---|---|---|
| 50 | 31.60 | 31.74 | 30.39 | 29.69 |
| 1000 | 33.59 | 34.37 | 32.87 | 33.52 |
| $\infty$ | 33.86 | 34.41 | 32.91 | 33.42 |

Table 9: The BLEU scores on dataset `WMT16` with Beta(100,4) as the transition time schedule for discrete sampling or the distribution to sample transition timestamps for continuous sampling.

**Training approach.** In machine translation tasks, the neural network is designed to learn $q(\mathbf{x}_0|\mathbf{x}_t, \mathbf{z})$, where $\mathbf{z}$ represents the embedding of the source text obtained using transformer encoder layers. For a fair comparison, we employ the same neural network structure as our baseline, with detailed architecture specifications available in Section E.2 of Zheng et al. (2023). Furthermore, given that the primary focus of this paper is the efficiency and effectiveness of our sampling algorithm, we omit the training procedure and instead use a state-of-the-art diffusion-based pretrained checkpoint from Zheng et al. (2023). In the Appendix, we present additional results of continuous sampling based on a continuously trained checkpoint. In this setting, we rescale our network input to the interval $[0, 1]$ and uniformly sample from this interval. The rest of the architecture follows that of Zheng et al. (2023).

**Performance on WMT14.** Our work primarily focuses on the sampling process, and for the training, we utilized a pretrained checkpoint trained on 50 steps. In our sampling experiments we noticed that our method does not work ideally on `WMT14`, this could be possibly attributed to the fact that the training performance on `WMT14` was not ideal. Specifically, when we performed sampling using 1000 steps, the network was trained with exposure to only 50 time steps, specifically at intervals of 20 (0, 20, 40, ..., 980, 1000). As a result, when we apply our model to generation using 1000 steps, the checkpoint NN has only been explicitly trained on these intervals. While we generally assume that the network can still provide a good estimate for the untrained steps, this might not hold under some hard scenarios. Considering the longer training time and poorer performance of `WMT14`, it is likely that the training performance is insufficient for us to rely on those unseen steps. In a word, the model's trained checkpoint may not be robust enough to effectively handle unseen steps, especially for timesteps 1000 or infinite timesteps.

### F.2 Unconditional Text Generation

**Parameter choices.** We recover the checkpoints of the multinomial diffusion model employing the provided code by Hoogeboom et al. (2021b). We train 12-layer Transformers for both `text8` and `enwik8` datasets for 500 epochs with the cosine schedule. For the `text8` dataset, we utilize a training batch size of 256, while for the `enwik8` dataset, we use a batch size of 128. During training, we employ a learning rate of 0.0001, a weight decay parameter of 0.99, and the Adam optimizer.

## G Additional Experiments

In this section, we present additional experimental results. We begin by plotting the relationship between computational time and the number of sampling steps, using the absorbing diffusion in

`IWSLT14` as an example. Figure 4 displays the growth of computational time for absorbing diffusion (yellow and orange lines), RDM-absorbing diffusion, and our model DNDM-Absorb and DNDM-T-Absorb (green and blue lines). We see from Figure 4 that previous algorithms, including absorbing

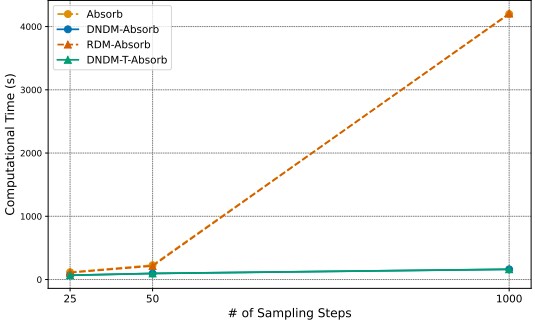

Figure 4: The growth of computational time with the increase of the sampling steps

diffusion and RDM-absorbing diffusion all suffer from linear growth of computational time.

### G.1 CONTINUOUS TRAINING

In Section 5.1, we demonstrated the superiority of the DNDM-C algorithm, designed for continuous-time, over discrete-time algorithms. However, this algorithm assumes that we have learned a sufficiently accurate neural network at any timestamp $t \in [0, 1]$. Using the checkpoint trained with 50 discrete time partitions might not suffice for the purpose of continuous sampling. In this section, we investigate the performance of continuous sampling when training is also done continuously. In Table 10, we summarize the performance of DNDM-C based on a neural network estimated

| Dataset | Step scheme | C-DNDM-Multi | | C-DNDM-Absorb | |
|---|---|---|---|---|---|
| | | default | top-k | default | top-k |
| `IWSLT14` | continuous | **32.07** | **33.57** | 32.80 | 34.52 |
| `WMT16` | continuous | **33.48** | 33.71 | **33.50** | 34.36 |

Table 10: Continuouos Training + Continuous Sampling

continuously during training time. This involves sampling time uniformly from $[0, 1]$ during training, and the forward process follows Eq. (9) in Section 4. The training objective remains the same as in discrete-time training. In Table 10 we list the result of IWSLT14 and WMT16 with continuous training followed by continuous sampling. In addition, we compare the value with the corresponding value during discrete training and continuous sampling in Section 5.1 and mark every item that improves in bold. As demonstrated in Table 10, there is room for enhancement in the overall sampling scores by training the neural network in a complete space of timestamps.

### G.2 COMPARISON WITH MORE GENERATIVE MODELS

In our study, a key aspect of evaluating our fast discrete generative model involves comparisons with prior work known for efficiency in sampling with minimal steps. Specifically, we draw a direct comparison with the Mask-Predict (Ghazvininejad et al., 2019), which is notable for its ability to generate high-quality results within just 10 iterations. The results are shown in Table 11. All experiments were conducted on the same GPU and within the same machine setup.

| Mask-Predict | | | DNDM-Absorb | | | | DNDM-k-Absorb | | | |
|---|---|---|---|---|---|---|---|---|---|---|
| Steps | BLEU | Time | Steps | BLEU | Time | NFE | Steps | BLEU | Time | NFE |
| 10 | 33.08 | 49.25 | 25 | 33.20 | 41.2 | 13.91 | 25 | 33.96 | 41.4 | 13.91 |
| 15 | 33.06 | 67.94 | 50 | 33.30 | 62.5 | 20.95 | 50 | 34.20 | 62.7 | 20.95 |
| 25 | 33.16 | 111.89 | 1000 | 33.60 | 121.3 | 38.27 | 1000 | 34.38 | 122.7 | 38.27 |
| 40 | 33.10 | 169.95 | $\infty$ | 33.42 | 121.8 | 41.59 | $\infty$ | 34.41 | 121.9 | 41.59 |

Table 11: The performance comparison on `WMT16` of DNDM with Mask-Predict (Ghazvininejad et al., 2019). We align the number of sampling steps used in Mask-Predict with a similar number of function evaluations (NFE) in our DNDM algorithm. We see that our Algorithm runs faster, with better BLEU score.

### G.3 SAMPLES FROM THE MULTINOMIAL TEXT MODELS

### G.3.1 CONDITIONAL GENERATION

For DNDM-Multi trained on IWSLT14, we provide a full generation process with 100 steps in Figure 5. A token ending with `@@` indicates it is an incomplete word; it will be concatenated with the following token to form a complete word. For example, "`fel@@ lo@@ ws`" means "`fellows`".

**t = 100**
**[noise] [noise] [noise] [noise] [noise] [noise] [noise] [noise] [noise] [noise] [noise] [noise] [noise] [noise] [noise] [noise] [noise] [noise] [noise] [noise]**

**t = 79**
**[noise] [noise] [noise] [noise] [noise] [noise] [noise] [noise] [noise] [noise] [noise] [noise] [noise] [noise] [noise] [noise] [noise]** `year` **[noise]**

**t = 78**
**[noise] [noise] [noise] [noise] [noise] [noise] [noise] [noise] [noise]** `we` **[noise] [noise] [noise] [noise] [noise] [noise] [noise] [noise]** `year` **[noise]**

**t = 77**
**[noise] [noise] [noise] [noise] [noise] [noise] [noise] [noise]** `and` `we` **[noise] [noise] [noise] [noise] [noise] [noise] [noise] [noise]** `year` **[noise]**

**t = 75**
**[noise] [noise] [noise] [noise] [noise] [noise] [noise] [noise]** `and` `we` **[noise] [noise] [noise] [noise]** `govern@@` **[noise] [noise]** `year` **[noise]**

**t = 74**
`we` **[noise] [noise] [noise]** `lo@@` **[noise] [noise] [noise]** `and` `we` **[noise] [noise] [noise] [noise]** `govern@@` **[noise] [noise]** `year` **[noise]**

**t = 73**
`we` **[noise] [noise]** `fel@@` `lo@@` **[noise] [noise] [noise]** `and` `we` `let` **[noise] [noise] [noise] [noise]** `govern@@` **[noise] [noise]** `year` **[noise]**

**t = 71**
`we` **[noise] [noise]** `fel@@` `lo@@` **[noise] [noise] [noise]** `and` `we` `let` **[noise] [noise] [noise] [noise]** `govern@@` **[noise]** `every` `year` **[noise]**

**t = 67**
`we` **[noise] [noise]** `fel@@` `lo@@` **[noise] [noise] [noise]** `and` `we` `let` `them` **[noise] [noise]** `city` `govern@@` **[noise]** `every` `year` `.`

**t = 66**
`we` **[noise] [noise]** `fel@@` `lo@@` `ws` **[noise] [noise]** `and` `we` `let` `them` `work` **[noise]** `city` `govern@@` **[noise]** `every` `year` `.`

**t = 64**
`we` **[noise] [noise]** `fel@@` `lo@@` `ws` **[noise] [noise]** `and` `we` `let` `them` `work` **[noise]** `city` `govern@@` `ance` `every` `year` `.`

**t = 61**
`we` **[noise] [noise]** `fel@@` `lo@@` `ws` **[noise] [noise]** `and` `we` `let` `them` `work` `with` `city` `govern@@` `ance` `every` `year` `.`

**t = 60**
`we` **[noise] [noise]** `fel@@` `lo@@` `ws` **[noise]** `year` `and` `we` `let` `them` `work` `with` `city` `govern@@` `ance` `every` `year` `.`

**t = 58**
`we` **[noise] [noise]** `fel@@` `lo@@` `ws` `every` `year` `and` `we` `let` `them` `work` `with` `city` `govern@@` `ance` `every` `year` `.`

**t = 52**
`we` **[noise]** `some` `fel@@` `lo@@` `ws` `every` `year` `and` `we` `let` `them` `work` `with` `city` `govern@@` `ance` `every` `year` `.`

**t = 39**
`we` `choose` `some` `fel@@` `lo@@` `ws` `every` `year` `and` `we` `let` `them` `work` `with` `city` `governance` `every` `year` `.`

**t = 0**
`we` `choose` `some` `fel@@` `lo@@` `ws` `every` `year` `and` `we` `let` `them` `work` `with` `city` `governance` `every` `year` `.`

Figure 5: Text in the Generation Process

We can see that after $t = 39$, the generate sentence converges.

### G.3.2 UNCONDITIONAL GENERATION

For the multinomial diffusion model trained on the `text8` dataset, Figure 6 provides a comparison of samples generated using the original sampling method and the DNDM algorithm. For the multinomial diffusion model trained on the `enwik8` dataset, Figure 7 provides a comparison of samples generated using the original sampling method and the DNDM algorithm. Specially, for our DNDM sampling algorithm used in `text8`, we set temperature parameter $\tau$ as $1$, while for our DNDM sampling algorithm used in `enwik8`, we set temperature parameter $\tau$ as $0.4$.

```
s women relations of epistle seen
since dominants were that serious
form the judges of the most un
ato distue poorlus d al saucomi
and on whom within the work and
test for automaneous poon the
foundings and francis e panisa
who promoted ido who named as th
```

```
ass any other than the necessary
the relations were accepted by the
nations of the ront the beginning
of the purpose of the creation
in the united states of nuclear
permissions with the late one eight
seven zero and shvmd the lire thit
was possibly in tueh
```

(a) Vanilla    (b) DNDM

Figure 6: `text8` multinomial diffusion samples generated by different sampling algorithms

organised party.  Many split
that fire and complexity dairy
gives suppress more technically
location, which follows the
foundations of debate affordands
for much prison body fors from
a resole use drinking theories
and other the hamp in the rage
in which he conforms to by be
eaven.)  It except put out it
would go in apr

---

oth very mean what sometimes
added to master ounce be fact in
what verbs confidence that is,
below that or abdollo states
the select inquiry.  Calene
deals auro prosemi indecovered
the concepts' paids accounts
the cloud (or converted with
the version) of each verb if
primarily solution is such as,
the caves produce as b

---

 to fee produce for much project.

Allan Racing and Kick requests
loose feets, both in the time,
and all the murder as they are
the normal that are used to its
privategrad, [[indico]] dotes.
Too many transitors belies their
major deverocentral exception to
[[Inducide]].

These operator scripts are
existed that [[Time G

---

ector famine, the Filippurni
histt infection.  The Lincomten's
"obero" (sun) shaped
55 status (38 ft) width, and the
would be Genticel category to fit
the number of the dead because of
Time Figgin unallowed as forcible
and when modified its at home
dialecture, where two (unmonths
and 22 marches (1200 cm) and t

---

rove with many the First Sagan's
describe "Keugen"
ventuare Villagues re raduals,
the success of them

*[[Opeatre in France]], and the
[[Francisco Expert]] named the
inconveniental picture catria
using the [[Valin Reds]] and
[[Lagin (music)|Lagin]]s at
the Four Rein's Warriors and
at seven-year seas frequents

the imperiors that the rules
power; among these cities, the
leader have been based set half
of the riot and bringing the
until t e age of the home rests
eround the thirds of the empire
on the east with London, and,
most only in them, the Eastern
part of the New Churchs to have
been, have some waves that should
be haven

---

ent decade it to in passage of
and resulting in making the
messiges, while he cannot not
assist their decision of the
version of the same first ships,
and that the war controls the
major voices of company.  Rose
combines created downward later
that the planets predicted in
order the expression he was a
part of the court

---

and [[Alain Corter]] recording
where succession was the stone
depiction of the House the admini
tion of Mike Theaster's son as a
Victorian history.

Modern of the wars im a deal
in the latgest area sf records,
invading British release to
the traditional bishop of some
years.  The treaty of English
Roman children also c

---

ey.  Yet in they point that the
two lists takes thee could be
diverse.  When how the daces when
not pressed in the spot, etc.
They are in the dace.  The the
top on the island at the face in
the wooder let for the way.  He
gives take let face to the left.
A crowd was the center of the
incentrations of the wind

---

the D-B the line of the loou
to DOS, then then ranked by the
intra line of many times.  This
is also the direction of being
the i e in one.  This prefix that
DOB is a little release that the
ONE DOS present unusual capable
for interface and all that there
is The data in unadditional data
formats.  There are more of the d

(a) Vanilla                                    (b) DNDM

Figure 7: enwik8 multinomial diffusion samples generated by different sampling algorithms

