# OpenReview forum: "Fast Sampling via De-randomization for Discrete Diffusion Models"
_ICLR.cc/2024/Conference — Submitted to ICLR 2024_

### Official Review · Reviewer_B4eP · 2023-10-15

**Soundness:** 3 good
**Presentation:** 3 good
**Contribution:** 2 fair
**Rating:** 5
**Confidence:** 4

**Summary:**

This paper presents an alternative formulation of the discrete multinomial-diffusion and absorbing-state-diffusion generative models, which is "de-randomized" in the sense that each token is randomly corrupted at most once during the forward noising process. This formulation motivates an "accelerated sampling" approach, in which they skip to the transition points instead of processing timesteps without token changes (although I'm not convinced this is done correctly). This allows them to use a finite number of generation steps for the continuous limit of infinite forward process steps. They find that their approach outperforms the "reparameterized diffusion model" (RDM) of Zheng et al. (2023) in terms of BLEU score for machine translation, and outperforms vanilla multinomial diffusion in terms of perplexity for text8 and enwiki8.

**Strengths:**

**[S1]** The alternative formulation of the forward process for multinomial diffusion and absorbing-state diffusion is interesting. This formulation, presented in Section 2.2, takes advantage of the fact that the noising distribution of these two types of discrete diffusion models is independent of the current state, e.g. the token at time $x_{t}$ is either exactly the same as $x_{t-1}$ or it is an independent draw from a noise distribution. This means you can construct a non-Markov process with the same conditional marginal distribution $q(x_t | x_0)$ by just picking a random time at which the token is replaced with a random token $w$, and then setting $x_t$ to either $w$ or $x_0$.

**[S2]** The proposed DNDM method seems to outperform RDM and vanilla multinomial diffusion for a fixed time budget, by skipping over wasted computation in RDM for large numbers of generation steps.

**[S3]** The paper is clearly written and easy to follow.

**Weaknesses:**

~~**[W1]**~~ *(addressed in revision)* I'm not convinced that the accelerated sampling algorithm that the authors propose is actually correct from a probabilistic standpoint. The authors motivate it by proving (in Theorem 3.1) that the transition times in the forward process are independent and follow a particular distribution based on the noise schedule. However, this is only proven for the *forward process*. In order for the proposed sampling algorithm to make sense, this property would have to also be true for the *reverse process*: it must be the case that, after conditioning on the current value of $x_t$, the transition times remain independent and have a tractable distribution.

This doesn't seem obvious to me. Note that the values of the tokens get noise added independently during the forward process when you condition on $x_0$, but are NOT independent when you condition on their values in $x_t$. This is why discrete diffusion methods are typically justified using a variational ELBO argument; approximating the tokens as independent creates a bias that must be corrected for through iterative refinement. The same might be true of the transition times.

**[W2]** *(mostly addressed in revision)* Overall, the proposed algorithm seems to rely on intuition and heuristics for some parts, and some important technical details about the probabilistic model are glossed over.

- ~~(a)~~ *(addressed in revision)* The paper refers to the predictive model of $p_\theta(x_{t-1} | \hat{x}_0, x_t)$ as a "score network". For continuous diffusion, the "score" refers to the gradient of the marginal density with respect to the input. I don't think this is quite correct in the discrete case. Prior work on discrete diffusion models mostly doesn't seem to use "score network" terminology, or else uses it to refer to a very specific generalization of the continuous score (e.g. [Meng et al. (2022)](https://arxiv.org/abs/2211.00802))
- ~~(b)~~ *(addressed in revision)* The assumptions being made about the "score network" are not clearly stated, although the paper later mentions "the possibility that WMT14 violates our assumption that the pre-trained score model is uniformly accurate for all time steps" in the experiments section. This seems important because the predictive model $p_\theta(x\_{t-1} | \hat{x}\_0, x\_t)$ is never perfectly accurate. In particular, in the standard diffusion model setup $p\_\theta$ makes independent predictions for all tokens, whereas those tokens may have a nontrivial joint distribution. Additionally it is only trained using a variational bound, so it seems likely to me that it would not be "uniformly accurate for all time steps".
- ~~(c)~~ *(addressed in revision)* Although the proposed DNDM model has an identical forward process, the corresponding approximate reverse process distribution isn't clearly explained; the authors instead jump straight to a sampling algorithm, and it's not obvious that optimizing the training objective will make this sampling algorithm match the forward process.
- (d) The authors note that their reverse sampler decodes predictions using "different decoding strategies including greedy decoding, top-k decoding and temperature decoding" and mentions that argmax decoding shows strong performance. I don't think this makes sense under a probabilistic view of the process; this feels just like a heuristic to improve results. *Edit: There's apparently some precent for this, since the RDM model also uses heuristic decoding methods.*

**[W3]** The authors motivate their approach based on accelerating generation speed, but I'm not convinced the gains are actually significant when compared to just using an autoregressive model. The authors seem to focus on accelerating sampling when there are more denoising steps than there are tokens (e.g. the 1000-step and $\infty$-step conditions for WMT, the 1000-step and 4000-step conditions for unconditional generation). But if you have more denoising steps than tokens, it seems like you may as well just use an ordinary autoregressive model. Indeed, in the "$\infty$-step" diffusion, the continuous-time reverse sampler requires one network evaluation per token generated, which is the same number of computation steps as an autoregressive model would use.

I expect that an autoregressive model would have a better tradeoff between generation time and quality than the method proposed here.

~~**[W4]**~~ *(addressed in revision)* The baselines for the experiments seem fairly limited. The authors only compare against RDM and vanilla multinomial diffusion models, but seem to introduce additional heuristics into the decoding process to get better results for a fixed compute budget. It seems to me then that it would be fairer to compare to the much larger set of non-autoregressive generative models, for which similar heuristics are considered fair game and for which many ideas have already been proposed and evaluated.

As one example, I believe Mask-Predict [(Ghazvininejad et al. 2019)](https://arxiv.org/abs/1904.09324) achieves higher BLEU scores on WMT14 (relative to this work) using only 10 model evaluations, by using argmax decoding and other heuristics. Many other examples are discussed in a survey by [Xiao et al. (2023)](https://arxiv.org/abs/2204.09269).

**[W5]** Unconditional generation samples (in the appendix) remain unimpressive relative to autoregressive model generation quality. This applies to previous work as well, of course, but I don't think the improvements in Table 3 translate to a noticeable increase in overall sample quality.

**[W6]** The proposed approach only applies to discrete diffusion models that have the form in Equation (1), but this is only a subset of the discrete diffusion models that have been considered in the literature. Austin et al. (2021) discusses a few alternatives that are not of this form (e.g. discreteized Gaussian, token-embedding-distance transitions, or hybrids of absorbing and other diffusion processes as discussed in the appendix). Another recent example is Blackout Diffusion [(Santos et al. 2023)](https://arxiv.org/abs/2305.11089).

**Questions:**

Can the proposed approach be formalized as a probabilistically-sound variational model, and if so, how? Relatedly, does the equivalent of Theorem 3.1 hold for the reverse process in general?

Relatedly, what assumptions are you imposing on the "score function"?

For the conditional generation tasks, I think the paper could be improved by including comparisons to ordinary autoregressive generation, and also to existing non-autoregressive machine translation baselines.

For the unconditional generation tasks, why don't you compare against RDM or absorbing diffusion? Also, how does your method do with a more realistic number of sampling steps (smaller than the sequence length), and how does it compare to an autoregressive model with the same compute budget?

I'm not sure what is meant by "full de-randomization". The transition times and the sampled tokens are still random, are they not?

Section 2.2 mentions "the chain rule" but it's not obvious to me what this is referring to. The equation seems correct but I don't think the derivation involves a chain rule (?)

---

> ### Author Response · Authors · 2023-11-22
> **Response to Reviewer B4eP (Part 1)**
>
> Thank you for your constructive feedback.
>
> **Q1-1.** Can the proposed approach be formalized as a probabilistically-sound variational model, and if so, how?
>
> **A1-1.** The proposed approach can indeed be formalized as a probabilistically-sound variational model. We have added the mathematical formulation in Appendix B.3 for our DNDM with both finite sampling steps and infinite sampling steps (i.e., continuous time sampling).
>
> ---
>
> **Q1-2.** I'm not convinced that the accelerated sampling algorithm that the authors propose is actually correct from a probabilistic standpoint. The authors motivate it by proving (in Theorem 3.1) that the transition times in the forward process are independent and follow a particular distribution based on the noise schedule. However, this is only proven for the forward process. … this property would have to also be true for the reverse process … after conditioning on the current value of, the transition times remain independent and have a tractable distribution.Relatedly, does the equivalent of Theorem 3.1 hold for the reverse process in general?
>
> **A1-2.** We believe there is a misunderstanding here by the reviewer.  Theorem 3.1 indeed holds for both the forward and reverse processes. More specifically, the independence of transition times $\tau_n$ follow from our definition of forward process in Eq. (4) and are not conditioned on $\mathbf{x}\_0$ or any other variables. In the forward process, we have $\mathbf{x}\_{t,n} = b_{t,n}\mathbf{x}\_{t-1,n} + (1-b_{t,n}) \mathbf{w}\_n $, and $\tau_{n} = t$ is equivalent to $b_{0,n} =1, \ldots, b_{t-1,n} = 1$ and $b_{t,n} = 0$. Since $\\{ b_{t,n} \\}\_{t=0}^T$ is independent for different $n$ by definition, each $\tau_n$ is also independent. Therefore, the corresponding reverse process also has independent transition time. We believe the reviewer's misunderstanding may stem from the pre-assumption that the transition of different tokens is dependent. However, we want to clarify that these transitions are, in fact, independent. This underscores the reason why transition time remains independent in both the forward and reverse processes.
>
> ---
>
> **Q2.**  The paper refers to the predictive model of $p_\theta\left( \mathbf{x}\_{t-1} \mid \hat{\mathbf{x}}\_0, x_t\right)$ as a "score network". For continuous diffusion, the "score" refers to the gradient of the marginal density with respect to the input. I don't think this is quite correct in the discrete case. Prior work on discrete diffusion models mostly doesn't seem to use "score network" terminology, or else uses it to refer to a very specific generalization of the continuous score
>
> **A2.** We apologize for the misleading use of words in our previous version. We have changed all “score network” to “neural network” in our revision.
>
> ---
>
> **Q3.** The assumptions being made about the "score network" are not clearly stated, although the paper later mentions "the possibility that WMT14 violates our assumption that the pre-trained score model is uniformly accurate for all time steps" in the experiments section. This seems important because the predictive model $p_\theta (\mathbf{x}\_{t-1} \mid \hat{\mathbf{x}}\_0, \mathbf{x}\_t )$ is never perfectly accurate. In particular, in the standard diffusion model setup $p_\theta$ makes independent predictions for all tokens, whereas those tokens may have a nontrivial joint distribution. Additionally it is only trained using a variational bound, so it seems likely to me that it would not be "uniformly accurate for all time steps". Relatedly, what assumptions are you imposing on the "score function"?
>
> **A3.** Thank you for raising this point. First of all, we don’t make any stronger assumption (e.g., perfectly accurate model) on the pre-trained model than existing works on diffusion models. Instead, we adopt the common practice of assuming the pre-trained model to be sufficiently effective, aligning with the approach taken by other works in the field.
>
> Second, in our paper, we employed a pretrained model checkpoint trained on 50 steps. This means the network was only familiarized with 50 distinct time steps during its training phase, and the timesteps during reverse sampling might have been mostly unseen, especially for timesteps 1000 or infinite timesteps. For example, the model trained on $t = 1, \ldots, 50$ might be asked to output at $t = 2.71$ during sampling. Therefore, we conjecture that the network can provide accurate estimates even for those steps which it was not explicitly trained on. However, for WMT14, the pretrained model checkpoint may not be able to generalize to unseen time steps effectively. We have now relocated this argument from the main text to an extended explanation in Appendix F.1 for clarity.
>
> ---

---

> ### Author Response · Authors · 2023-11-22
> **Response to Reviewer B4eP (Part 2)**
>
> **Q4.** Although the proposed DNDM model has an identical forward process, the corresponding approximate reverse process distribution isn't clearly explained; the authors instead jump straight to a sampling algorithm, and it's not obvious that optimizing the training objective will make this sampling algorithm match the forward process.
>
> **A4.** We believe there is a misunderstanding regarding the reverse process by the reviewer. The identical forward process indicates an identical reverse process distribution. The approximate error does not come from the non-accurate reverse process distribution but from the approximation error of training the neural network.
>
> To connect the training objective and our DNDM, we have added the training objective and the ELBO in Appendix B.3. Our analysis shows that optimizing the training objective will make this sampling algorithm match the forward process and this training objective aligns with previous discrete diffusion models up to a constant.
>
> ---
>
> **Q5.** The authors note that their reverse sampler decodes predictions using "different decoding strategies including greedy decoding, top-k decoding and temperature decoding" and mentions that argmax decoding shows strong performance. I don't think this makes sense under a probabilistic view of the process; this feels just like a heuristic to improve results.
>
> **A5.** This is again a misunderstanding regarding the decoding strategy and decoding type. The argmax decoding in our algorithm is different from the decoding strategies mentioned in "different decoding strategies including greedy decoding, top-k decoding and temperature decoding", where the latter refers to the decoding strategy in a traditional NLP task aimed to get $\hat{\mathbf{x}}\_0$, such as the decoding strategy used in Transformer. In contrast, the decoding type in Section 5 refers to the decoding scheme used in $p(\mathbf{x}\_{t-1} | \mathbf{x}\_t, \hat{\mathbf{x}}\_0)$ in the reverse sampling process.
>
> ---

---

> ### Author Response · Authors · 2023-11-22
> **Response to Reviewer B4eP (Part 3)**
>
> **Q6.** The authors motivate their approach based on accelerating generation speed, but I'm not convinced the gains are actually significant when compared to just using an autoregressive model. The authors seem to focus on accelerating sampling when there are more denoising steps than there are tokens (e.g. the 1000-step and $\infty$-step conditions for WMT, the 1000-step and 4000-step conditions for unconditional generation). But if you have more denoising steps than tokens, it seems like you may as well just use an ordinary autoregressive model. Indeed, in the "$\infty$-step" diffusion, the continuous-time reverse sampler requires one network evaluation per token generated, which is the same number of computation steps as an autoregressive model would use.
> I expect that an autoregressive model would have a better tradeoff between generation time and quality than the method proposed here.
>
> **A6.** We respectfully disagree that our focus is on accelerating sampling when there are more denoising steps $T$ than the number of tokens $N$. While it is true that our method can get more significant improvement when T is larger, we care about the result for all sampling steps, including those much smaller than the number of tokens. Below is an example of our result shown in Table 5 of our revision.
>
> | Dataset | Steps | RDM-Multi |  | DNDM-Multi |  | RDM- $k$-Multi |  | DNDM- $k$-Multi |  |
> | :---: | :---: | :---: | :---: | :---: | :---: | :---: | :---: | :---: | :---: |
> |  |  | BLEU | Avg NFE | BLEU | Avg NFE | BLEU | Avg NFE | BLEU | Avg NFE |
> | IWSLT14 | 25 | **31.26** | 25 | 30.95 | **9.03** | **32.82** | 25 | 32.30 | **9.03** |
> |  | 50 | **31.50** | 50 | 31.45 | **14.07** | **32.82** | 50 | 32.80 | **14.07** |
> |  | 1000 | 31.69 | 1000 | **31.82** | **30.33** | 32.64 | 1000 | **33.15** | **30.33** |
> |  | $\infty$ | - | - | **31.89** | **32.73** | - | - | **33.44** | **32.73** |
> | WMT14 | 25 | **25.25** | 25 | 25.01 | **13.52** | **26.03** | 25 | 25.98 | **13.52** |
> |  | 50 | **25.75** | 50 | 25.33 | **20.58** | 26.14 | 50 | **26.37** | **20.58** |
> |  | 1000 | 25.66 | 1000 | **25.71** | **38.94** | 25.82 | 1000 | **26.88** | **38.94** |
> |  | $\infty$ | - | - | **24.79** | **40.67** | - | - | **26.39** | **40.67** |
> | WMT16 | 25 | **32.29** | 25 | 31.97 | **8.5** | **33.12** | 25 | 32.94 | **8.5** |
> |  | 50 | **32.53** | 50 | 32.50 | **14.73** | **33.41** | 50 | 33.26 | **14.73** |
> |  | 1000 | 32.63 | 1000 | **32.86** | **38.45** | 33.67 | 1000 | **33.79** | **38.45** |
> |  | $\infty$ | - | - | **32.91** | **41.64** | - | - | **33.86** | **41.64** |
>
>
>
> As we can see from the table, our algorithm is 2-3 times faster even for small sampling steps (25). We have also proved that our algorithm can accelerate the discrete diffusion for all $N, T$ and included more discussion in Appendix D.
>
> In addition, we agree that our infinite limit has the same NFE as the autoregressive diffusion. However, our method also supports finite step sampling, which is much faster than the autoregression diffusion in terms of NFE. Moreover, our primary objective and contribution revolve around expediting multinomial diffusion-type discrete diffusions. The comparison with autoregressive models does not contribute to substantiating our assertion regarding the acceleration of sampling speed.
>
> ---

---

> ### Author Response · Authors · 2023-11-22
> **Response to Reviewer B4eP (Part 4)**
>
> **Q7.** The baselines for the experiments seem fairly limited. The authors only compare against RDM and vanilla multinomial diffusion models, but seem to introduce additional heuristics into the decoding process to get better results for a fixed compute budget. It seems to me then that it would be fairer to compare to the much larger set of non-autoregressive generative models, for which similar heuristics are considered fair game and for which many ideas have already been proposed and evaluated. As one example, I believe Mask-Predict (Ghazvininejad et al. 2019) achieves higher BLEU scores on WMT14 (relative to this work) using only 10 model evaluations, by using argmax decoding and other heuristics. Many other examples are discussed in a survey by Xiao et al. (2023).
>
> **A7.** We would like to emphasize that the main contribution of our work is to accelerate the sampling speed of multinomial diffusion-type discrete diffusion models. So the most important and relevant baselines are multinomial diffusion and absorbing diffusion *without* using our sampling technique, i.e., RDM and vanilla multinomial/absorbing diffusion.
>
> Furthermore, we don’t introduce additional heuristics. In our decoding process, we strictly adhere to the same mechanisms employed by the algorithms we compare with, specifically those in RDM. The introduction of various decoding types in our study is not an arbitrary heuristic but rather a collection of existing decoding strategies used in RDM. This approach exactly ensures that our comparison with RDM is fair.
>
> As per your recommendation, we have included a comparison with Mask-Predict, although it's important to note that our primary contribution does not center around surpassing Mask-Predict.  A summary of these results is provided below for reference. On the WMT16 dataset, our model demonstrates uniformly superior performance than Mask-Predict.
>
> On the WMT14 dataset, we encountered challenges in replicating the results reported in Mask-Predict due to a staled link of the data (This is also reported as an issue in their Github repo). We were able to run the algorithm but the performance does not match. Instead, we reference the result of 27.03 BLEU score reported in their paper. Our model, DNDM-k-Absorb, achieves a BLEU score of 27.18, which is better than their result. This performance was attained within 68.0 seconds, which is on par with 67.15 seconds reported for Mask-Predict on the same machine.
>
> The performance comparison on WMT16 of DNDM with Mask-Predict
> | | | Mask-Predict     |  | |   DNDM-Absorb |  |  | |   DNDM-k-Absorb |  |  |
> | :---: | :---: | :---: | :---: | :---: | :---: | :---: | :---: | :---: | :---: | :---: | :---: |
> | Steps | BLEU | Time | Steps | BLEU | Time | NFE | Steps | BLEU | Time | NFE |  |
> | 10 | 33.08 | 49.25 | 25 | 33.20 | 41.2 | 13.91 | 25 | 33.96 | 41.4 | 13.91 |  |
> | 15 | 33.06 | 67.94 | 50 | 33.30 | 62.5 | 20.95 | 50 | 34.20 | 62.7 | 20.95 |  |
> | 25 | 33.16 | 111.89 | 1000 | 33.60 | 121.3 | 38.27 | 1000 | 34.38 | 122.7 | 38.27 |  |
> | 40 | 33.10 | 169.95 | $\infty$ | 33.42 | 121.8 | 41.59 | $\infty$ | 34.41 | 121.9 | 41.59 |  |
>
>
> The performance comparison on WMT14 of DNDM with Mask-Predict
> | | | Mask-Predict   |   | |  DNDM-Absorb  |  |  | | DNDM-k-Absorb   |  |  |
> | :---: | :---: | :---: | :---: | :---: | :---: | :---: | :---: | :---: | :---: | :---: | :---: |
> | Steps | BLEU | Time | Steps | BLEU | Time | NFE | Steps | BLEU | Time | NFE |  |
> | 10 | 27.03 | 67.15 | 25 | 25.79 | 68.1 | 15.09 | 25 | 27.18 | 68.0 | 15.09 |  |
> | 15 | -  | 95.24 | 50 | 26.10 | 102.0 | 22.45 | 50 | 27.66 | 102.5 | 22.45 |  |
> | 25 | -  | 154.21 | 1000 | 26.43 | 178.3 | 38.76 | 1000 | 27.82 | 179.1 | 38.76 |  |
>
> ---

---

> ### Author Response · Authors · 2023-11-22
> **Response to Reviewer B4eP (Part 5)**
>
> **Q8.** Unconditional generation samples (in the appendix) remain unimpressive relative to autoregressive model generation quality. This applies to previous work as well, of course, but I don't think the improvements in Table 3 translate to a noticeable increase in overall sample quality.
>
> **A8.** Again, our contribution is on the sampling speed instead of the sample quality. Since our method is a training-free accelerated sampling algorithm, the performance of our method in terms of sample quality is contingent on the pre-trained models, which may not always be optimized for sample quality.
>
> ---
>
> **Q9.** The proposed approach only applies to discrete diffusion models that have the form in Equation (1), but this is only a subset of the discrete diffusion models that have been considered in the literature. Austin et al. (2021) discusses a few alternatives that are not of this form (e.g. discreteized Gaussian, token-embedding-distance transitions, or hybrids of absorbing and other diffusion processes as discussed in the appendix). Another recent example is Blackout Diffusion (Santos et al. 2023).
>
> **A9.** Eq. (1) has already included the most two widely used discrete diffusion models:  multinomial diffusion and absorbing diffusions. Note that the discretized Gaussian diffusion and Blackout diffusion only outperform other discrete diffusion models for continuous data, while we focus on the generation of discrete data such as text. We leave the acceleration of other “discrete” diffusion models as a future work.
>
> ---
>
> **Q10.** For the unconditional generation tasks, why don't you compare against RDM or absorbing diffusion? Also, how does your method do with a more realistic number of sampling steps (smaller than the sequence length), and how does it compare to an autoregressive model with the same compute budget?
>
> **A10.** Based on our experiments, none of the discrete diffusion models exhibited satisfactory performance for unconditional generation tasks. As a result, we have opted to exclusively present the results of multinomial diffusion and its accelerated version to provide readers with a representative overview of these outcomes.
>
> ---
>
> **Q11.** I'm not sure what is meant by "full de-randomization". The transition times and the sampled tokens are still random, are they not?
>
> **A11.** The concept of 'full de-randomization' is introduced to contrast with the discrete diffusion model proposed by Song et al. (2020a). In their model, each $\mathbf{x}\_{t-1}$ relies on $\mathbf{x}\_0$, $\mathbf{x}\_t$, and an injected noise term $Cat(1_K)$, as detailed in Equation (18) of Song et al. (2020a). In contrast, our reverse process depends on $\mathbf{x}\_0$, $\mathbf{x}\_t$, and a random transition time $\tau$, but notably, it does not incorporate any injected noise (as all previous discrete diffusion models do). From the perspective of given $\mathbf{x}\_0$, $\mathbf{x}\_t$, and transition time, our approach achieves full de-randomization.
>
> ---
>
> **Q12.** Section 2.2 mentions "the chain rule" but it's not obvious to me what this is referring to. The equation seems correct but I don't think the derivation involves a chain rule (?)
>
> **A12.** Thank you for pointing out, this is indeed a typo. We were trying to explain how one can get $p(\mathbf{x}\_t | \mathbf{x}\_0)$ from the Markov chain characterized by $p(\mathbf{x}\_t | \mathbf{x}\_{t-1})$ and marginal distribution. We have revised our paper accordingly.
>
> ---
> [1] Ghazvininejad, Marjan, et al. "Mask-predict: Parallel decoding of conditional masked language models." arXiv preprint arXiv:1904.09324 (2019).
>
> [2] Lin, Xiao, et al. "Discrete Conditional Diffusion for Reranking in Recommendation." arXiv preprint arXiv:2308.06982 (2023).
>
> [3] Song, Jiaming, Chenlin Meng, and Stefano Ermon. "Denoising diffusion implicit models." ICLR 2021.
>
> [4] Austin, Jacob, et al. "Structured denoising diffusion models in discrete state-spaces." Advances in Neural Information Processing Systems 34 (2021): 17981-17993.

---

> ### Author Response · Authors · 2023-11-23
> **Looking forward to your feedback**
>
> Dear Reviewer B4eP,
>
> We are grateful for your valuable feedback. In our rebuttal, we addressed your concerns, clarified misunderstandings, and incorporated additional theoretical and empirical results as per your suggestions. As the discussion deadline is approaching, we would like to hear your feedback and address any further questions you may have. Thank you!
>
> Best regards,
> Authors

---

> > ### Comment · Reviewer_B4eP · 2023-11-23
> > **Discussion**
> >
> > Due to this rebuttal being posted very close to the end of the discussion period, I have not yet had a chance to fully read the revised submission or respond to everything in the rebuttal, but I wanted to quickly reply with follow up comments and things I still do not find clear about the response.
> >
> > **A1-1**: Yes, I am aware that the transition times are independent as defined under the forward process. However, it is possible for two random variables to be independent overall and yet become dependent when conditioning on a third variable. The reverse process generally involves conditioning on the value of the partially-denoised sequence, and my concern was that the transition times might become conditionally dependent at this point even though they are independent overall.
> >
> > Looking at the ELBO in the appendix, however, it seems that you always sample the transition times $\tau$ first, and never infer them from a partially-denoised sequence? In this case this may not be an issue.
> >
> > **A4** I disagree that this is a misunderstanding. Training a diffusion model generally involves fitting an approximate posterior distribution that is less expressive than the true posterior, e.g. by assuming that the posterior factorizes in a particular way and thus that certain variables are conditionally independent. Your model appears to be implicitly using a different factorization of the approximate posterior than previous work, and my concern was that the constraints you are imposing didn't seem clearly explained. (I have not yet had a chance to carefully look at Appendix B, which may answer my question.)
> >
> > I also want to point out that the choice of how the approximate reverse process (or more generally an approximate posterior) factorizes can directly introduce approximation error if the conditional independence relationships are incorrect, independent of the error involved with fitting the neural network.
> >
> > **A5** Your answer here doesn't make sense to me. Section 3 is describing your method, isn't it? So aren't these decoding strategies also part of your method?
> >
> > Regardless, using argmax decoding in Section 5 seems somewhat problematic from a probabilistic standpoint on its own, since this is no longer sampling from the probabilistic model that was used to build the ELBO.
> >
> > **A7** Thanks for clarifying that RDM also uses other decoding strategies such as argmax decoding, which is surprising to me.

---

> > > ### Author Response · Authors · 2023-11-23
> > > **Response to Additional Comments**
> > >
> > > Thank you for your additional questions. We answer them as follows.
> > >
> > > ---
> > >
> > > “Follow-up A1-1: Yes, I am aware that the transition times are independent as defined under the forward process. However, it is possible for two random variables to be independent overall and yet become dependent when conditioning on a third variable. The reverse process generally involves conditioning on the value of the partially-denoised sequence, and my concern was that the transition times might become conditionally dependent at this point even though they are independent overall.
> > > Looking at the ELBO in the appendix, however, it seems that you always sample the transition times $\tau$ first, and never infer them from a partially-denoised sequence? In this case this may not be an issue.”
> > >
> > > **Answer**: Yes, we sample the transition times $\tau$ first rather than inferring them from a partially denoised sequence, which is shown both in Line 3 of Algorithm 1, and the ELBO we added in Appendix B. Therefore, $\tau$’s are always i.i.d. and probabilistically sound. We hope this will totally remove your concern.
> > >
> > > ---
> > >
> > > “Follow-up A4: I disagree that this is a misunderstanding. Training a diffusion model generally involves fitting an approximate posterior distribution that is less expressive than the true posterior, e.g. by assuming that the posterior factorizes in a particular way and thus that certain variables are conditionally independent. Your model appears to be implicitly using a different factorization of the approximate posterior than previous work, and my concern was that the constraints you are imposing didn't seem clearly explained. (I have not yet had a chance to carefully look at Appendix B, which may answer my question.)
> > >
> > > I also want to point out that the choice of how the approximate reverse process (or more generally an approximate posterior) factorizes can directly introduce approximation error if the conditional independence relationships are incorrect, independent of the error involved with fitting the neural network.”
> > >
> > > **Answer**: In Section B.3, which was added in the revision, we have explicitly articulated our approach to factorize the posterior distribution, and this factorization leads to the iterates of our reverse sampling process. More specifically, for the exact reverse process defined by $q(\boldsymbol{x}\_{t-1} | \boldsymbol{x}\_t, \boldsymbol{x}\_0, \tau)$, since $\boldsymbol{x}\_{t-1}$ solely depends on  $\boldsymbol{x}\_0, \tau, \boldsymbol{x}\_t$, given $\boldsymbol{x}\_0$ and $\tau$, the reverse process exhibits the Markovian property. Consequently, the factorization $ q(\boldsymbol{x}\_{1:T}|\boldsymbol{x}\_0,\tau) = q(\boldsymbol{x}\_T | \boldsymbol{x}\_0, \tau) \Pi_{t=2}^T q(\boldsymbol{x}\_{t-1} | \boldsymbol{x}\_{t}, \boldsymbol{x}\_{0}, \tau)$ holds. On the other hand, for the approximate reverse process defined by $p_{\theta}(\boldsymbol{x}\_{t-1}|\boldsymbol{x}\_t, \tau)$, we utilize $p_\theta(\boldsymbol{x}\_{0:T}|\tau)  =  p_\theta^{(T)}(\boldsymbol{x}\_T | \tau) \prod_{t=1}^T p_\theta^{(t)}(\boldsymbol{x}\_{t-1} | \boldsymbol{x}\_{t}, \tau)$ and this formulation has been explained in detail in Section B.3 of the revision.
> > >
> > > ---
> > >
> > > “ Follow-up A5. Your answer here doesn't make sense to me. Section 3 is describing your method, isn't it? So aren't these decoding strategies also part of your method?
> > >
> > > Regardless, using argmax decoding in Section 5 seems somewhat problematic from a probabilistic standpoint on its own, since this is no longer sampling from the probabilistic model that was used to build the ELBO.”
> > >
> > >
> > > **Answer**: For the decoding strategy in Section 3, it is a part of $f_{\theta}(\boldsymbol{x}\_{t,1:N}, t)$, similar to the output layer of the neural network. Instead of being associated with the diffusion model, it is more pertinent to the NLP task. In essence, it is orthogonal to all discrete diffusion models. The argmax decoding mentioned in “decoding type” paragraph in Section 5 also refers to the decoding strategy related to the $\hat{x}_{0, n}$ network and is therefore irrelevant to our method. The argmax decoding strategy has also been introduced in Section 5.3 and detailed in Appendix E.4 of Zheng et al. (2023).
> > >
> > > For the top-k-selection related to our diffusion model, in Appendix E, we have attributed this to RDM (Zheng et al. 2023), specifically the “improved decoding” in Section 5.3 and detailed in Appendix E.4 of Zheng et al. (2023). This is to ensure a fair comparison with RDM.
> > >
> > > ---
> > >
> > > Zheng, L., Yuan, J., Yu, L., & Kong, L. (2023). A reparameterized discrete diffusion model for text generation. arXiv preprint arXiv:2302.05737.
> > >
> > > ---
> > >
> > > Please let us know if you have any further questions. Thank you!

---

### Official Review · Reviewer_yTYV · 2023-10-27

**Soundness:** 3 good
**Presentation:** 2 fair
**Contribution:** 2 fair
**Rating:** 5
**Confidence:** 4

**Summary:**

The paper presents an accelerated sampling method for discrete diffusion models by transforming traditional discrete diffusion forward processes to more deterministic equivalents, so that only a single transition is made in the forward process for each independent token. Sampling is then done by sampling the transition times for each token (along with the prior tokens), and iterating over each step that has any transitions occurring in it. Since for large step counts, there may be some time steps with no transitions occurring at all, the proposed method accelerates sampling by skipping over them. The paper further goes to present an infinite-step limit, where each token gets a different transition time, providing a maximum to the time required for sampling. The paper goes on to showcase that the method achieves competitive results in machine translation compared to another method for accelerating discrete diffusion sampling, and also provides clear improvements over standard discrete diffusion models in unconditional text generation on small-scale data sets.

**Strengths:**

+ The topic is important due to the slowness of most current discrete diffusion models. While there is quite a bit of work on accelerating continuous diffusion model sampling, less work exists on discrete diffusion model acceleration.
+ I think the idea is very sensible and presented in a very understandable manner.
+ Does provide improvements over baseline D3PM model, as well as the previous RDM method with relatively large step counts
+ I found it especially surprising that the new sampling procedure outperformed the baseline uniform diffusion model on text8 so clearly, since seems that cutting down the sampling steps in a smart way actually improves generation quality. This is quite counterintuitive, given that usually increasing steps improves results in diffusion models.

**Weaknesses:**

- I think that the use of the phrase “sampling iterations” in the paper can be somewhat misleading, for instance on page 5, where it is indicated that the generation time increases sub-linearly with sampling iterations (as opposed to RDM). While technically true from the diffusion perspective, this also overlooks the fact that the relevant metrics are actually the amount of function evaluations (or the time taken for sampling). Similarly, comparisons with the same step counts to RDM in Table 1 & 2 seem suboptimal to me, and instead it should be (approximately) the same amount of function evaluations / wall clock time.
- It would also be good to elaborate where exactly does the method provide an improvement over RDM, since they focused more on step counts <25, whereas the comparisons here are for step counts >25. In this sense, is the model even a direct competitor for RDM, or could the two methods be combined?
- I think the paper may require more work on clarifying what are the new contributions of the proposed model and designing the experiments such that they clearly showcase these contributions (see also questions on ARDM). E.g., under what conditions exactly can we expect improvements in sampling speed?

**Questions:**

- It seems to me that there are quite a bit of (potentially subtle) similarities with the previous work [1], where the authors note a connection between absorbing diffusion models and order-agnostic autoregressive models. In particular, as pointed out in their Appendix C, the continuous-time limit of an absorbing diffusion model is effectively equivalent to to their order-agnostic ARDM, which seems to be also equivalent to the infinite-time limit for absorbing diffusion presented in this paper. Would the authors of the submission agree, or maybe are there differences that I have overlooked? I do see that in the non-infinite-time case, the models do not exactly match, and the model is formulated from a different point of view, potentially allowing more flexibility in designing the generative process. But it would be good to give more elaboration on the differences to ARDM and maybe also provide direct numerical comparisons in the paper.
- One downside of the paper as of now is that a method to estimate the data likelihood is not included. Perhaps it would be possible to create a lower bound by not having having intermediate latents $x_t$ at all, and considering the transition times $\tau$ as latent variables? For instance, setting the inference distribution to $q(\tau^{1:D}) \prod_k^D q(x_T^k | x_0^k)$, where $\tau^k$ is distributed according to Uniform({1,…,T}), and the corresponding generative process defined such that we first sample from the priors $p(x_T)$ and $p(\tau)$, and then generate the different dimensions according to the ordering of $\tau$. If I am not mistaken, this would be similar to what was done in the ARDM paper as well, and one could form an ELBO similar to theirs.
- How is the evaluation with the BERT model done for the unconditional generation experiments, exactly?

Overall, I think the idea is good, but if I am not mistaken on the points regarding novelty compared to ARDM, the paper requires more work on finding and clearly presenting the unique angle in which the new discrete diffusion formulation contributes to the community. Thus, I think that the paper is not ready for publication yet.

References:
[1]: Autoregressive Diffusion Models. Emiel Hoogeboom, Alexey A. Gritsenko, Jasmijn Bastings, Ben Poole, Rianne van den Berg, Tim Salimans. ICLR 2022.

---

> ### Author Response · Authors · 2023-11-22
> **Response to Reviewer yTYV (Part 1)**
>
> Thank you for your helpful comments！
>
> ---
>
> **Q1**. I think that the use of the phrase “sampling iterations” in the paper can be somewhat misleading, for instance on page 5, where it is indicated that the generation time increases sub-linearly with sampling iterations (as opposed to RDM). While technically true from the diffusion perspective, this also overlooks the fact that the relevant metrics are actually the amount of function evaluations (or the time taken for sampling). Similarly, comparisons with the same step counts to RDM in Table 1 & 2 seem suboptimal to me, and instead it should be (approximately) the same amount of function evaluations / wall clock time.
>
>
> **A1**.  Thank you for your suggestion. We have used “sampling steps” to replace “sampling iterations”  in the revision. While the number of sampling steps is also a widely used metric as used in [1] and [2], we have also incorporated the NFE metric into our experiments to provide a more comprehensive comparison in our revision.  Please refer to Tables 5 and 6 in Appendix D, where we list the average NFEs of both RDM and our method, under the same sampling steps. Below is an example of our result shown in Table 6 of our revision. We can see that the BLEU score achieved by our method with about 40 NFE outperforms the BLEU score of RDM with 50 NFE.
>
>
> | Dataset | | RDM-Absorb |  | DNDM-Absorb |  | RDM- $k$-Absorb |  | DNDM- $k$-Absorb |  |
> | :---: | :---: | :---: | :---: | :---: | :---: | :---: | :---: | :---: | :---: |
> |  |  Steps | BLEU | Steps | BLEU | Avg NFE | BLEU | Avg NFE | BLEU | Avg NFE |
> | IWSLT14 | 25 | 31.58 | 25 | **32.43** | **13.81** | **34.50** | 25 | 34.14 | **13.81** |
> |  | 50 | 31.80 | 50 | **32.63** | **19.24** | **34.58** | 50 | 34.34 | **19.24** |
> |  | 1000 | 31.91 | 1000 | **32.93** | **31.08** | **34.60** | 1000 | 34.56 | **31.08** |
> |  | &infin; | - | - | **33.03** | **32.07** | - | - | **34.65** | **32.07** |
> | WMT14 | 25 | 24.97 | 25 | **25.79** | **15.09** | **27.50** | 25 | 27.18 | **15.09** |
> |  | 50 | 24.95 | 50 | **26.10** | **22.45** | **27.73** | 50 | 27.66 | **22.45** |
> |  | 1000 | 25.22 | 1000 | **26.43** | **38.76** | 27.75 | 1000 | **27.82** | **38.76** |
> |  | &infin; | - | - | **26.50** | **40.39** | - | - | **27.50** | **40.39** |
> | WMT16 | 25 | 32.86 | 25 | **33.20** | **13.91** | 33.92 | 25 | **33.96** | **13.91** |
> |  | 50 | 32.93 | 50 | **33.30** | **20.95** | 34.10 | 50 | **34.20** | **20.95** |
> |  | 1000 | 33.25 | 1000 | **33.60** | **38.27** | **34.44** | 1000 | 34.38 | **38.27** |
> |  | &infin; | - | - | **33.42** | **41.59** | - | - | **34.41** | **41.59** |
>
> ---
> **Q2**. It would also be good to elaborate where exactly does the method provide an improvement over RDM, since they focused more on step counts <25, whereas the comparisons here are for step counts >25. In this sense, is the model even a direct competitor for RDM, or could the two methods be combined?
>
> **A2**. Our model is not intended as a direct competitor to RDM but rather as a training-free accelerator in terms of sampling speed, which can be integrated with various discrete diffusion models in the form of (1), including RDM. Our approach addresses a key inefficiency inherent in existing sampling algorithms of discrete diffusion models. Specifically, in traditional diffusion models, including RDM, increasing the number of sampling steps leads to better sampling quality but at the cost of linearly increasing computational time, often with only marginal performance gains. In contrast, our method achieves a more favorable tradeoff between sampling time and quality. The improvement in efficiency is because our model significantly reduces the number of function evaluations (NFE), as our method. We have added Tables 5 and 6 in Appendix D with NFE as a performance metric.  We can see that even for 1000 sampling steps, the effective number of sampling steps (NFE) is around 30-40. Therefore, with smaller NFE (30-40), our method can consistently achieve a higher BLEU score than RDM with 50 NFE. Please see the answer to A1 and Tables 5 and 6 in our revision for the detailed comparison in terms of NFE.

---

> ### Author Response · Authors · 2023-11-22
> **Response to Reviewer yTYV (Part 2)**
>
> **Q3**. I think the paper may require more work on clarifying what are the new contributions of the proposed model and designing the experiments such that they clearly showcase these contributions (see also questions on ARDM). E.g., under what conditions exactly can we expect improvements in sampling speed?
>
> **A3**. As we emphasized in the introduction, our main contribution is proposing a new sampling method for discrete diffusion models that reduces the generation time (i.e., NFE) while preserving the sample quality.  Please see the table in A1 for details.
>
> From the theoretical point of view, for discrete diffusion models, our method is guaranteed to achieve a reduction in NFE compared with the number of sampling steps. This is proved in Theorem D.1 in the newly added Appendix D. More specifically, Theorem D.1 shows that the NFE is reduced from $T$ to $(1- C_{D_{\tau}, T, N})T$, where $C_{D_{\tau}, T, N}$ is a constant with range $(0,1)$ that depend on $D_{\tau}, T, N$.  The details of the discussion can be found in Appendix D.
>
> ---
>
> **Q4**. It seems to me that there are quite a bit of (potentially subtle) similarities with the previous work, where the authors note a connection between absorbing diffusion models and order-agnostic autoregressive models. In particular, as pointed out in their Appendix C, the continuous-time limit of an absorbing diffusion model is effectively equivalent to their order-agnostic ARDM, which seems to be also equivalent to the infinite-time limit for absorbing diffusion presented in this paper. Would the authors of the submission agree, or maybe are there differences that I have overlooked? I do see that in the non-infinite-time case, the models do not exactly match, and the model is formulated from a different point of view, potentially allowing more flexibility in designing the generative process. But it would be good to give more elaboration on the differences to ARDM and maybe also provide direct numerical comparisons in the paper.
>
> **A4**. Thank you for bringing our attention to the ARDM. We would like to emphasize that our method is a training-free fast sampling method for various discrete diffusion processes, including absorbing diffusion and multinomial diffusion. Such a fast sampling method enables us to tackle discrete diffusion models with a large number of sampling steps, or even infinite steps. While ARDM can be seen as the infinite-time limit of absorbing diffusion, it remains orthogonal to our training-free fast sampling method.
>
> ---
>
> **Q5**. One downside of the paper as of now is that a method to estimate the data likelihood is not included. Perhaps it would be possible to create a lower bound by not having intermediate latents \(x_t\) at all, and considering the transition times \(\tau\) as latent variables? For instance, setting the inference distribution to \(q(\tau_{1:T})\prod_{t=1}^k q(x_t^k|x_t^0)\), where \(x_t^k\) is distributed according to Uniform(\{1,...,T\}), and the corresponding generative process defined such that we first sample from the priors \(p(x_T)\) and \(p(\tau)\), and then generate the different dimensions according to the ordering of \(\tau\). If I am not mistaken, this would be similar to what was done in the ARDM paper as well, and one could form an ELBO similar to theirs.
>
> **A5**. Thank you for your valuable advice. We have provided the derivation of ELBO in Appendix B.3 which can estimate the data likelihood. The ARDM obtains their ELBO by decomposing with respect to the data dimension, which depends on the autoregressive nature of their model. Therefore, their method doesn’t apply to our model. Instead, we take the approach of classic diffusion models, especially the non-markovian continuous time diffusion model DDIM [3], and derive the ELBO for our model.  Our ELBO is based on both intermediate latent $x_t$ and the transition time $\tau$.

---

> ### Author Response · Authors · 2023-11-22
> **Response to Reviewer yTYV (Part 3)**
>
> **Q6**. How is the evaluation with the BERT model done for the unconditional generation experiments, exactly?
>
> **A6**. Thanks for pointing this out. We realized that BERT may not be a good model for evaluating the perplexity of unconditionally generated texts. We have reevaluated the perplexity scores (PPL) of the generated samples using the GPT2-large model. Our new evaluation results are as follows: the average perplexity score for the text samples generated from multinomial diffusion trained on enwik8 using vanilla sampling methods is 801.7817. Additionally, the average perplexity score for the text samples generated from multinomial diffusion trained on the text8 dataset via DNDM is 556.7812.
>
> ---
> [1] Austin, Jacob, et al. "Structured denoising diffusion models in discrete state-spaces." Advances in Neural Information Processing Systems 34 (2021): 17981-17993.
>
> [2] Zheng, Lin, et al. "A reparameterized discrete diffusion model for text generation." arXiv preprint arXiv:2302.05737 (2023).
>
> [3] Song, Jiaming, Chenlin Meng, and Stefano Ermon. "Denoising diffusion implicit models." arXiv preprint arXiv:2010.02502 (2020).

---

> > ### Comment · Reviewer_yTYV · 2023-11-22
> > **Response to rebuttal**
> >
> > I thank the authors for the answers and the new content!
> >
> > I think that the new text and points made do make the paper more convincing. However, I would still like to raise some discussion on the connection to ARDM. In the infinite-step limit, in practice the generation seems to decompose to the same thing: Since there is zero probability that two transitions are set on the same time, the generation ends up going through each data dimension separately, similarly to ARDM. The main difference seems to be that with a finite amount of steps, there is some probability that the two transitions coincide. I do acknowledge that it could be valuable to have a well-defined framework to interpolate between these two extremes of "multiple tokens are changed at each step" (small amount of steps, no/small difference to standard discrete diffusion) and the other extreme of "only one token is changed at each step" (infinite steps, small difference to ARDM?). In addition, it could be valuable to have such a framework defined from the point of view of the more standard discrete diffusion framework instead of the ARDM framework, where the connection requires more elaboration. Conceptually, applying the method presented in the paper to a discrete diffusion model could be easier to researchers working with discrete diffusion models than switching the perspective to the ARDM framework. In any case, it seems to me still that discussion and positioning regards to it should be made clear in the paper.
> >
> > It is also worth pointing out that the ARDM paper also considers parallelizing the generation to fewer than N steps. Also, while ARDM does not apply to other forward processes than absorbing, it is also the case that here the other processes than absorbing do not exactly correspond to their original counterparts: In uniform diffusion, it is possible for a token to make many transitions during the diffusion process all the way to the end, while here the accelerated uniform diffusion seems to work closer to absorbing diffusion, where only one step is made for each token.
> >
> > I wonder if the authors agree on these points, or perhaps I have misrepresented something?

---

> ### Author Response · Authors · 2023-11-23
> **Response to Additional Comments**
>
> Thank you very much for your positive feedback! We would like to clarify and elaborate more on the above points.
>
> ---
>
> “ In the infinite-step limit, in practice the generation seems to decompose to the same thing: Since there is zero probability that two transitions are set on the same time, the generation ends up going through each data dimension separately, similarly to ARDM. ”
>
> **Answer**: We partially agree. Note that in the infinite time limit ($N$ is fixed while $T \rightarrow \infty$), if the transition time distribution concentrates on several different points, there can still be several transitions happening at the same time. For example,  let us consider a setting with two tokens. Assume the transition distribution for each token is the same: with probability $1/2$, $\tau = 1$, and with probability $1/2$, $\tau$ is uniform on $[0,1]$. In this case, the transition time of these two tokens can still be the same with probability 0.5*0.5 = 1/4, which is not zero probability.
>
> ---
>
> “In any case, it seems to me still that discussion and positioning regards to it should be made clear in the paper”
>
> **Answer**: Thank you for your suggestion.  For our method, when $N$ is fixed while $T \rightarrow \infty$, the total NFE will reach $N$.  On the other hand, when $T$ is fixed and $N \rightarrow \infty$, the NFE will reach $T$. Such observation can be rigorously inferred from our Theorem D.1. Therefore, our framework bridges two extremes of (fixed N, infinite T) and (infinite N, finite T). We summarize the NFE of our method under different regimes in the table below. As you can see, our method can accelerate sampling for both finite time-steps and infinite time. In contrast, ARDM is primarily focused on infinite timesteps. We have added a discussion in Section 3 in the revision.
>
>
> | NFE| $N$ | $N \rightarrow \infty$ |
> |----------|----------|----------|
> | $T$   | $(1 - C_{N,T,D_{\tau}})\cdot T$    | $T$     |
> | $T \rightarrow \infty$    | $N $   |  Intractable |
>
> ---
>
> "It is also worth pointing out that the ARDM paper also considers parallelizing the generation to fewer than N steps. "
>
> **Answer**: Thank you for your suggestion. For infinite timesteps, ARDM proposed an advanced parallelizing technique that can reduce NFE according to the log-likelihood, which we haven’t considered in our DNDM-C. We have added it in Section 4.1 in the revision.
>
> ---
>
> “Also, while ARDM does not apply to other forward processes than absorbing, it is also the case that here the other processes than absorbing do not exactly correspond to their original counterparts: In uniform diffusion, it is possible for a token to make many transitions during the diffusion process all the way to the end, while here the accelerated uniform diffusion seems to work closer to absorbing diffusion, where only one step is made for each token.”
>
> **Answer**: We agree that in multinomial diffusion, it is possible for a token to make many transitions during the diffusion process all the way to the end during the forward process in Eq. (1) in our paper. However, we want to clarify that using our de-randomized technique, we can derive a new process in Eq. (4), where each token only has one transition. This new process has the same conditional probability flow $q(\boldsymbol{x}\_{t}|\boldsymbol{x}\_{0})$ as the original multinomial diffusion. We have also derived an ELBO for Eq. (4) and proved its equivalence to the original process in Section B.3.
>
> ---
>
> To sum up, we have added a new paragraph in Section 4 to discuss the ARDM. You can find it in the revised paper.

---

### Official Review · Reviewer_y1n6 · 2023-10-30

**Soundness:** 3 good
**Presentation:** 3 good
**Contribution:** 3 good
**Rating:** 6
**Confidence:** 3

**Summary:**

The authors propose a de-randomized diffusion method that can be used to accelerate existing discrete diffusion processes while preserving the conditional distributions. From here, they propose DNDM, a discrete non-Markov diffusion model, and evaluate the approach on language generation and machine translation tasks. The technique allows for a continuous-time sampling algorithm, which is demonstrated to have superior performance than discrete-time finite-step counterparts.

**Strengths:**

Originality:
- The paper proposes a de-randomization procedure for discrete diffusion processes, which allows for a huge boost to sampling speed. Furthermore, the procedure allows for a continuous-time formulation of the discrete diffusion process, which the authors demonstrate outperforms the finite-time counterparts.

Quality:
- The method is well-justified, and the mathematical framework behind the de-randomization process is clearly formulated. The experimental results demonstrate a clear speedup to sampling speed, as well as non-trivial boosts to BLEU score on conditional text generation tasks.

Clarity:
- The paper is generally well-written and clear.

Significance:
- Since the proposed de-randomization procedure is orthogonal to existing discrete diffusion processes (e.g. can be applied out of the box to Multinomial Diffusion (1) and Absorbing Diffusion (2)), and leads to sampling speedups and non-trivial performance gains, this work is likely relevant to many researchers in the area of discrete diffusion.

1. Emiel Hoogeboom, Didrik Nielsen, Priyank Jaini, Patrick Forré, and Max Welling. Argmax flows and multinomial diffusion: Learning categorical distributions.
2. Jacob Austin, Daniel D Johnson, Jonathan Ho, Daniel Tarlow, and Rianne Van Den Berg. Structured denoising diffusion models in discrete state-spaces.

**Weaknesses:**

- The experiments are promising, but the scope is a bit limited. For example, the claims of the paper could be strengthened with results on sequence-to-sequence generation tasks such as question generation and paraphrasing (1, 2).
- An ablation of the choice to use Beta distribution to approximate the true transition time distribution (induced by the $\alpha_t$ schedule during training) would be clarifying. This is especially true in the case of the continuous time evaluation (Tables 1, 2), where it's slightly unclear whether the boosts to performance are due to the continuous time formulation or the choice of Beta distribution.

1. Lin Zheng, Jianbo Yuan, Lei Yu, and Lingpeng Kong. A reparameterized discrete diffusion model for text generation.
2. Gong, S., Li, M., Feng, J., Wu, Z., and Kong, L. Diffuseq: Sequence to sequence text generation with diffusion models.

**Questions:**

Questions:
- The derivation of the de-randomization process (e.g. Thm 2.1, Thm 3.1) shows the conditional distributions $q(x_t \mid x_0)$ should be consistent with the standard Markov diffusion process. Then, what is the explanation for why DNDM generally underperforms RDM with 25 to 50 steps on conditional text generation (Tables 1, 2)?
- Similarly, why is it that DNDM consistently outperforms RDM at 1000 steps (Tables 1, 2)? Is this due to variance reduction in the sampling process with high step number, or the approximation of the true transition time distribution using the Beta distribution (or something else)?
- How well does the model compare to RDM when the Beta distribution transition times are not used?

Clarification:
- Regarding the WMT14 results, the authors mention the weaker performance relative to the other datasets could be a sign that WMT14 "violates our assumption that the pretrained score model is uniformly accurate for all time steps" -- could you clarify what this means?

---

> ### Author Response · Authors · 2023-11-22
> **Response to Reviewer y1n6 (Part 1)**
>
> Thank you for your strong support!
>
> **Q1**. The experiments are promising, but the scope is a bit limited. For example, the claims of the paper could be strengthened with results on sequence-to-sequence generation tasks such as question generation and paraphrasing (1, 2).
>
>
> **A1**. Thank you for your valuable suggestion. We have added Theorem D.1, which shows that the NFE of our method is provably reduced from $T$ to $(1- C_{D_{\tau}, T, N})T$. Therefore, our method is guaranteed to achieve a reduction in NFE compared with the number of sampling steps. We're currently in the middle of conducting experiments on sequence-to-sequence generation tasks. We will add these additional experiments to the final version.
>
> ---
>
> **Q2**. An ablation of the choice to use Beta distribution to approximate the true transition time distribution (induced by the $\( \alpha_t \) $schedule during training) would be clarifying. This is especially true in the case of the continuous time evaluation (Tables 1, 2), where it's slightly unclear whether the boosts to performance are due to the continuous time formulation or the choice of Beta distribution.
>
> **A2**. Thank you for your valuable suggestion. We have conducted an ablation study on different choices of Beta distributions (Tables 7, 8, 9 in Appendix F.1). Specifically, when utilizing the same Beta distribution as those employed in continuous time, the discrete (finite step) version of DNDM does not achieve comparable performance as DNDM-C. Please refer to the following table and Table 9 in Appendix F.1.
>
> BLEU scores of different sampling steps with the same transition time distribution $Beta(100, 4)$ on WMT16:
>
> | Steps | DNDM-k-multi | DNDM-k-absorb | DNDM-multi | DNDM-absorb |
> | :---: | :--- | :--- | :--- | :--- |
> | 50 | 31.60 | 31.74 | 30.39 | 29.69 |
> | 1000 | 33.59 | 34.37 | 32.87 | 33.52 |
> | $\infty$ | 33.86 | 34.41 | 32.91 | 33.42 |
>
> We note that, in the last column, although the performance of 1000 steps is slightly better than infinite steps, this aligns with the result of DNDM-absorb in Table 2, and therefore does not contradict our conclusion that the difference in Beta-distribution used does not significantly impact the performance comparison between 1000 steps and infinite steps. This suggests that the performance improvements are mainly attributed to the continuous time formulation itself.
>
> ---
> **Q3**. How well does the model compare to RDM when the Beta distribution transition times are not used?
>
> **A3**. We appreciate your insightful question. As detailed in the ablation study in Appendix C, we conducted experiments using various transition time distributions. The effect of different distributions can be seen as different schedules for selecting $\alpha_t$ in discrete diffusion models. In particular, In Table 4, $cosine^2$ schedule corresponds to the $\alpha_t$ schedule utilized in the RDM results. Overall, we observed a slight decrease in performance when not employing the Beta distribution. However, our approach still demonstrates superior performance compared to both RDM-absorbing and RDM-multinomial with comparable amounts of NFEs. Please refer to the following table and Table 4 in Appendix C for more details.
>
> | Datasets | Schedules | DNDM-multi |  | DNDM-absorb |  | DNDM-k-multi |  | DNDM-k-absorb |  |
> | :--- | :--- | :--- | :--- | :--- | :--- | :--- | :--- | :--- | :--- |
> |  |  | BLEU | Avg NFE | BLEU | Avg NFE | BLEU | Avg NFE | BLEU | Avg NFE |
> | IWSLT14 | Cosine | 31.72 | 31.71 | 32.71 | 31.21 | 32.91 | 31.71 | 34.50 | 31.21 |
> |  | Cosine $^2$ | 31.78 | 31.74 | **32.93** | 31.21 | 32.78 | 31.74 | 34.53 | 31.21 |
> |  | Linear $\alpha$ | 31.77 | 31.82 | 32.65 | 31.33 | 32.83 | 31.82 | 34.53 | 31.33 |
> |  | Beta (reported) | **31.82** | **30.33** | **32.93** | **31.08** | **33.15** | **30.33** | **34.56** | **31.08** |
> | WMT14 | Cosine | **25.80** | 39.61 | **26.54** | 39.18 | 26.63 | 39.61 | 27.81 | 39.18 |
> |  | Cosine ${ }^2$ | 25.52 | 39.48 | 26.53 | 39.18 | 25.01 | 39.48 | **27.95** | 39.18 |
> |  | Linear $\alpha$ | 25.58 | 39.97 | 26.33 | 39.82 | 25.47 | 39.97 | 27.63 | 39.82 |
> |  | Beta (reported) | 25.71 | **38.94** | 26.43 | **38.76** | **26.88** | **38.94** | 27.82 | **38.76** |
> | WMT16  | Cosine | 32.71 | 40.50 | 33.56 | 40.45 | 33.46 | 40.50 | 34.37 | 40.45 |
> |  | Cosine ${ }^2$ | 32.73 | 40.50 | 33.51 | 40.45 | 33.44 | 40.50 | 34.24 | 40.45 |
> |  | Linear $\alpha$ | 32.85 | 40.36 | 33.46 | 40.36 | 33.47 | 40.36 | 33.88 | 40.36 |
> |  | Beta (reported) | **32.86** | **38.46** | **33.60** | **38.27** | **33.79** | **38.45** | **34.38** | **38.27** |

---

> ### Author Response · Authors · 2023-11-22
> **Response to Reviewer y1n6 (Part 2)**
>
> **Q5**. Regarding the WMT14 results, the authors mention the weaker performance relative to the other datasets could be a sign that WMT14 "violates our assumption that the pretrained score model is uniformly accurate for all time steps" -- could you clarify what this means?
>
> **A5**. Thank you for pointing this out. We apologize for any confusion caused by our initial argument.
> In our paper, we employed a pretrained model checkpoint trained on 50 steps. This means the network was only familiarized with 50 distinct time steps during its training phase, and the timesteps during reverse sampling might have been mostly unseen, especially for timesteps 1000 or infinite timesteps. For example, the model trained on $t = 1, \ldots, 50$ might be asked to output at $t = 2.71$ during sampling. Therefore, we conjecture that the network can provide accurate estimates even for those steps which it was not explicitly trained on. However, for WMT14, the pretrained model checkpoint may not be able to generalize to unseen time steps effectively. We have now relocated this argument from the main text to an extended explanation in Appendix F.1 for clarity.

---

> > ### Comment · Reviewer_y1n6 · 2023-11-23
> >
> > Thanks for the response. The additional experimental results are helpful and compelling. I believe the discussion with Reviewer yTYV has also been valuable and they raise important points about comparison to ARDM. The recent revisions from the authors is appreciated but a more thorough section positioning this work with respect to existing methods would be helpful. Overall I will keep my score as is.

---

> > > ### Author Response · Authors · 2023-11-23
> > >
> > > Thank you for your affirmative feedback. We have added paragraphs in Sections 3 and 4 to compare with ARDM in the revision as suggested by Reviewer yTYV. We really appreciate your continued support!

---

### Official Review · Reviewer_7d5g · 2023-11-09

**Soundness:** 3 good
**Presentation:** 3 good
**Contribution:** 3 good
**Rating:** 6
**Confidence:** 5

**Summary:**

Very interesting work. Assuming a diffusion process to exists several transition times ---when the process will be still after that time. Using this nice property can achieve acceleration of sampling steps. The authors study both the discrete process and the continuous process. The authors first point out that for each process there is a transition time. Then they prove that transition time are independently spread and follows a certain analytic density. Following this density, the authors design algorithms to jump at each transition time to achieve acceleration of diffusion sampling. Their method demonstrates efficiency in several scenarios using small NFEs.

**Strengths:**

This paper is novel, in the asense it proposes to leverage the transition time in the diffusion process to accelerate the sampling. As far as I know, no one before thought of that. And this method indeed achieves nice performance.

**Weaknesses:**

This paper is not complete. First of all, it lacks ablation studies to demonstrate the influence of distributions of the transition time. With respect, I do not agree with that, all instances of transition times are equal in effect.

Following up on the above point, why not the author study which instantiation of the transition time is the best for sampling, in the sense of quality and speed? Stopping at the distribution stage is not a good thought, I believe.

Another thing is the visualization of transition times, the x_t at those times, and other related values. Letting me see that x_t gets still after the transition time will make me more convinced of your method and creativity.

Apart from that, the authors did not implement this work in image generation. I guess they may have a good reason for that, as diffusion is the dominating and most influential method in image generation. A method to improve diffusion sampling but not implemented on the image is strange. Are there anything barriers to the method to apply in images?

Also, I believe the authors also agree with, the proposed method does not outperform competitors in all shown cases. Sometimes the improvements are marginal.

As a theoretical work, I appreciate the idea a lot, but the theoretical analysis is not that deep so I can safely vote for acceptance with no objection from the other reviewers and the AC. Deeper theoretical analysis related to sampling NFEs, approximation error with the original process, and techniques to further improve it could be much more appreciated.

I am not an expert in the language generation domain, if I have any mistakes, do please correct me.

The major problem of this paper to me, is the lack of enough empirical analysis, and results on image domains.

**Questions:**

Algorithms like DPM-Solvers do not work in the language case? Please kindly tell me.

---

> ### Author Response · Authors · 2023-11-22
> **Response to Reviewer 7d5g (Part 1)**
>
> Thank you very much for your support! Below we address the questions.
>
> ----
>
> **Q1**. This paper is not complete. First of all, it lacks ablation studies to demonstrate the influence of distributions of the transition time. With respect, I do not agree with that, all instances of transition times are equal in effect.
>
> **A1**. Thank you for your suggestion. We have added several ablation studies regarding this matter. Specifically, in Table 4 of our revision, we present a comparison of the effects by different schedules of transition time on our algorithm's performance. For reference, we also provide the table of our ablation study here. We can see that our proposed Beta schedule outperforms previous schedules.
>
> | Datasets | Schedules | DNDM-multi |  | DNDM-absorb |  | DNDM-k-multi |  | DNDM-k-absorb |  |
> | :--- | :--- | :--- | :--- | :--- | :--- | :--- | :--- | :--- | :--- |
> |  |  | BLEU | Avg NFE | BLEU | Avg NFE | BLEU | Avg NFE | BLEU | Avg NFE |
> | IWSLT14 | Cosine | 31.72 | 31.71 | 32.71 | 31.21 | 32.91 | 31.71 | 34.50 | 31.21 |
> |  | Cosine $^2$ | 31.78 | 31.74 | **32.93** | 31.21 | 32.78 | 31.74 | 34.53 | 31.21 |
> |  | Linear $\alpha$ | 31.77 | 31.82 | 32.65 | 31.33 | 32.83 | 31.82 | 34.53 | 31.33 |
> |  | Beta (reported) | **31.82** | **30.33** | **32.93** | **31.08** | **33.15** | **30.33** | **34.56** | **31.08** |
> | WMT14 | Cosine | **25.80** | 39.61 | **26.54** | 39.18 | 26.63 | 39.61 | 27.81 | 39.18 |
> |  | Cosine ${ }^2$ | 25.52 | 39.48 | 26.53 | 39.18 | 25.01 | 39.48 | **27.95** | 39.18 |
> |  | Linear $\alpha$ | 25.58 | 39.97 | 26.33 | 39.82 | 25.47 | 39.97 | 27.63 | 39.82 |
> |  | Beta (reported) | 25.71 | **38.94** | 26.43 | **38.76** | **26.88** | **38.94** | 27.82 | **38.76** |
> | WMT16 | Cosine | 32.71 | 40.50 | 33.56 | 40.45 | 33.46 | 40.50 | 34.37 | 40.45 |
> |  | Cosine ${ }^2$ | 32.73 | 40.50 | 33.51 | 40.45 | 33.44 | 40.50 | 34.24 | 40.45 |
> |  | Linear $\alpha$ | 32.85 | 40.36 | 33.46 | 40.36 | 33.47 | 40.36 | 33.88 | 40.36 |
> |  | Beta (reported) | **32.86** | **38.46** | **33.60** | **38.27** | **33.79** | **38.45** | **34.38** | **38.27** |
>
> Additionally, Tables 7, 8, and 9 in our revision are dedicated to comparing the performance under various parameter choices in the Beta distribution.
>
> ----
>
> **Q2**. Following up on the above point, why not the author study which instantiation of the transition time is the best for sampling, in the sense of quality and speed? Stopping at the distribution stage is not a good thought, I believe.
>
> **A2**. If we understand your question correctly, it refers to a specific set of choices of transition times. The exploration of different transition times presents several practical challenges. First of all, the number of transition times equals N, corresponds to the length of a sentence, which is not fixed. Secondly, for long sequences with N=30, the total number of possible transition times is $T^{30} \geq 2^{30}$ . It is thus impractical to enumerate every possible instantiation. Therefore, we have focused our efforts on comparing the performance impacts of different distributions of transition times. Please see A1 for more details.
>
> ----
>
> **Q3**. Another thing is the visualization of transition times, the $x_t$ at those times, and other related values. Letting me see that $x_t$ gets still after the transition time will make me more convinced of your method and creativity.
>
> **A3**.   In Figure 5 in Appendix G.3.1, we have included an entire generation trajectory of $x_t$ with respect to the transition time of each token. This trajectory shows how $x_t$ evolves over time following their respective transition times. In particular, $x_t$ gets still after $t=39$. We also provide a simplified trajectory in Figure 2(b).
>
> ----
>
> **Q4**. Apart from that, the authors did not implement this work in image generation. I guess they may have a good reason for that, as diffusion is the dominating and most influential method in image generation. A method to improve diffusion sampling but not implemented on the image is strange. Are there anything barriers to the method to apply in images?
>
> **A4**. Our paper primarily focuses on the acceleration of sampling for discrete diffusion models on discrete space, especially on models in the form of (1) in our paper, which includes multinomial diffusion [1] and absorbing diffusion [2]. Image generation, on the contrary, typically falls into the realm of continuous diffusion models due to the continuous nature of the image data. According to [1] and [3], the performance of discrete diffusion models for image generation, even the more sophisticated discretized Gaussian [2], or blackout models [3], is worse than the continuous diffusion models such as DDPM [4]. Therefore, we choose to prioritize evaluation of our models on discrete data generation tasks, as we don’t think discrete diffusion models can easily beat DDPM on image generation.

---

> ### Author Response · Authors · 2023-11-22
> **Response to Reviewer 7d5g (Part 2)**
>
> **Q5**. Also, I believe the authors also agree with, the proposed method does not outperform competitors in all shown cases. Sometimes the improvements are marginal.
>
> **A5**. We agree. However, we want to clarify that the primary contribution of our method lies in its accelerated sampling speed. We use the term *outperform* to specifically refer to the enhanced sampling speed achieved by our method, as opposed to improvement in the overall generation performance (e.g., BLEU score). Instead, we show that under the same number of sampling steps, our approach attains a competitive BLEU score as RDM, with substantially faster generation. For example, for 1000 steps, RDM on WMT16 with k-absorb is slightly better than our 1000 steps, but is 20 times slower than our approach.
>
> ----
>
> **Q6**. As a theoretical work, I appreciate the idea a lot, but the theoretical analysis is not that deep so I can safely vote for acceptance with no objection from the other reviewers and the AC. Deeper theoretical analysis related to sampling NFEs, approximation error with the original process, and techniques to further improve it could be much more appreciated.
>
> **A6**. Thank you for your suggestion. In the appendix, we have added an analysis on the number of function evaluations (NFE). Specifically, in Theorem D.1,  we have theoretically established that the NFE in our method will not exceed $(1 - (1 - \frac{1}{T})^{N})\cdot T$, as opposed to $T$  NFE for the original sampling method. This suggests that our method is more efficient in terms of NFE. Detailed proofs and discussions regarding NFE can be found in Appendix D.
>
> Regarding the approximation error to the original process, we would like to clarify that there is no approximation error.  In Appendix B.1,  we have shown that our new process is equivalent to the original process in Eq.(1).
>
> ----
>
> **Q7**. Algorithms like DPM-Solvers do not work in the language case? Please kindly tell me.
>
> **A7**. The DPM-Solver [5] is exclusively designed for continuous diffusion applications. At the core of DPM-Solver is a specialized high-order ordinary differential equation (ODE) solver tailored for the semi-linear structure of continuous diffusion models (e.g., DDPM, VP-SDE and VE-SDE), employing methods such as Taylor expansions. It is unclear if DPM-Solver can be applied to discrete diffusion.
>
> On the other hand, a lot of recent studies [6,7,8] have shown that continuous diffusion models do not work as well as GPT or discrete diffusion models for language generation.
>
> ----
>
> [1] Hoogeboom, Emiel, et al. "Argmax flows and multinomial diffusion: Learning categorical distributions." Advances in Neural Information Processing Systems 34 (2021): 12454-12465.
>
> [2] Austin, Jacob, et al. "Structured denoising diffusion models in discrete state-spaces." Advances in Neural Information Processing Systems 34 (2021): 17981-17993.
>
> [3] Santos, Javier E., et al. "Blackout Diffusion: Generative Diffusion Models in Discrete-State Spaces." arXiv preprint arXiv:2305.11089 (2023).
>
> [4] Ho, Jonathan, Ajay Jain, and Pieter Abbeel. "Denoising diffusion probabilistic models." Advances in neural information processing systems 33 (2020): 6840-6851.
>
> [5] Lu, Cheng, et al. "Dpm-solver: A fast ode solver for diffusion probabilistic model sampling in around 10 steps." Advances in Neural Information Processing Systems 35 (2022): 5775-5787.
>
> [6] Li, Xiang, et al. "Diffusion-lm improves controllable text generation." Advances in Neural Information Processing Systems 35 (2022): 4328-4343.
>
> [7] Gong, Shansan, et al. "Diffuseq: Sequence to sequence text generation with diffusion models." arXiv preprint arXiv:2210.08933 (2022).
>
> [8] Ye, Jiasheng, et al. "Diffusion Language Models Can Perform Many Tasks with Scaling and Instruction-Finetuning." arXiv preprint arXiv:2308.12219 (2023).

---

> ### Author Response · Authors · 2023-11-22
> **Thank you**
>
> Dear Reviewer 7d5g,
>     Thank you for increasing your score. We're glad that our rebuttal and revision have addressed your questions.
> Authors

---

> > ### Comment · Reviewer_7d5g · 2023-11-23
> >
> > Please make sure you add those components into the paper in this final revision. Especially ablation studies.
> >
> > Also, note to add a comma after equations, there are components of sentences.

---

> > > ### Author Response · Authors · 2023-11-23
> > >
> > > Thank you for your encouraging feedback. We have added the missing punctuations for  some equations, and also added all the ablation studies in Appendix C of the revision. We look forward to your continued support!

---

### Meta-Review · Area_Chair_SC3o · 2023-12-09

**Metareview:**

The AC would like to thank both the authors and reviewers for engaging in constructive discussions. During the rebuttal and the discussion phase, misunderstandings were resolved, additional probabilistic interpretations were included and additional results in the form of an ablation study and a comparison against mask-predict were added. This had led to three out of four reviewers raising their score. However, even with those raised scores, the reviews remain mixed with half of the reviewers scoring the paper as marginally below the acceptance threshold, and the other half as marginally above. The main concern that remains is the significance of the proposed methods compared to related work such as ARDMs by Hoogeboom et al. 2022 and tau-leaping accelerated sampling by Campbell et al 2022. Although the reviewers and authors have discussed the conceptual comparison against ARDMS, and the authors have included new paragraphs in the revision to compare and contrast the current work with ARDMS conceptually, two reviewers remain unconvinced that the current exposition of this submission demonstrates a significant enough contribution compared to this work. Furthermore, during the reviewer-AC discussion tau-leaping accelerated sampling was also mentioned as a method that requires a more in-depth comparison. Taking this into account, the AC recommends rejection of this paper for this conference, and encourages the authors to more clearly contrast and compare their work to related work, both conceptually and through empirical experiments, and to resubmit this improved version to another venue.

**Justification For Why Not Higher Score:**

Two out of four reviewers have indicated that the contributions in this paper are insufficiently distinguished from related work, making it hard to judge its merit.

**Justification For Why Not Lower Score:**

n/a

---

### Decision · Program_Chairs · 2024-01-16

Reject